

# Studying catchment storm response using event and pre-event water volumes as fractions of precipitation rather than discharge

Jana von Freyberg[1,2], Bjørn Studer[1], Michael Rinderer[3], James W. Kirchner[1,2]

[1]Department of Environmental Systems Science, ETH Zurich, Zurich, Switzerland
[2] Swiss Federal Institute for Forest, Snow and Landscape Research (WSL), Birmensdorf, Switzerland
[3]Chair of Hydrology, University of Freiburg, Freiburg im Breisgau, Germany

*Correspondence to*: Jana von Freyberg (jana.vonfreyberg@usys.ethz.ch)

**Abstract**

Streamflow response to precipitation is often investigated using isotope-based hydrograph separation, which quantifies the contribution of precipitation (i.e., event water $Q_e$) or water from storage (i.e., pre-event water $Q_{pe}$) to total discharge ($Q$) during storm events. In order to better understand streamflow generating mechanisms, hydrograph separation studies often seek to relate the event water fraction $Q_e/Q$ to storm characteristics or antecedent wetness conditions. However, these relationships may be obscured because the same factors that influence $Q_e$ also necessarily influence total discharge $Q$ as well. Here we propose that the fractions of event water and pre-event water relative to precipitation instead of discharge ($Q_e/P$ and $Q_{pe}/P$) provide useful alternative tool for studying catchment storm responses. These two quantities separate the well-known runoff coefficient ($Q/P$, i.e. the ratio between discharge and precipitation over the event time scale) into its contributions from event water and pre-event water. Whereas the runoff coefficient $Q/P$ quantifies how strongly precipitation inputs affect streamflow, the fractions $Q_e/P$ and $Q_{pe}/P$ track the sources of this streamflow response.

We use high-frequency measurements of stable water isotopes for 24 storm events at a steep headwater catchment (Erlenbach, central Switzerland) to compare the storm-to-storm variations in $Q_e/Q$, $Q_e/P$ and $Q_{pe}/P$. Our analysis explores how storm characteristics and antecedent wetness conditions affect the mobilization of event water and pre-event water at the catchment scale. Isotopic hydrograph separation shows that streamflow was typically dominated by pre-event water, although event water exceeded 50% of discharge for several storms. No clear relationships were found linking either storm characteristics or antecedent wetness conditions with the volumes of event water or pre-event water ($Q_e$, $Q_{pe}$), or with event water as a fraction of discharge ($Q_e/Q$), beyond the unsurprising correlation of larger storms with greater $Q_e$ and greater total $Q$. By contrast, event water as a fraction of precipitation ($Q_e/P$) was strongly correlated with storm volume and intensity but not with antecedent wetness, implying that the volume of event water that is transmitted to streamflow increases more-than-





proportionally with storm size under both wet and dry conditions. Conversely, pre-event water as a fraction of precipitation ($Q_{pe}/P$) was strongly correlated with all measures of antecedent wetness but not with storm characteristics, implying that wet conditions primarily facilitate the mobilization of old (pre-event) water, rather than the fast transmission of new (event) water to streamflow, even at a catchment where runoff coefficients can be large.

Thus, expressing event and pre-event water volumes as fractions of precipitation rather than discharge was more insightful for investigating the Erlenbach catchment's hydrological behaviour. If $Q_e/P$ and $Q_{pe}/P$ exhibit similar relationships with storm characteristics and antecedent wetness conditions in other catchments, we suggest that these patterns may potentially be useful as diagnostic "fingerprints" of catchment storm response.

**1 Introduction**

Studying catchment hydrological responses to precipitation events can be useful in identifying dominant controls on streamflow generation. For decades, hydrologists have compared the volumes of precipitation ($P$) and discharge ($Q$) during storm events using the runoff coefficient $Q/P$ (e.g., Fischer et al., 2017; Horton, 1933; Jordan, 1994; Litt et al., 2015; McGlynn et al., 2004; Munyaneza et al., 2012; Shanley and Chalmers, 1999; Sidle et al., 1995). Other studies have separated the hydrograph into baseflow and quickflow (using graphical methods, low-pass digital filtering, or recession

curve analysis; Blume et al., 2007; Hall, 1968; Hewlett and Hibbert, 1967; Nathan and McMahon, 1990), and compared the quickflow fraction of $Q$ to precipitation. Both the runoff coefficient $Q/P$ and the ratio of quickflow to precipitation quantify how streamflow responds to precipitation inputs, but neither tracks the source of this streamflow response. In other words, both metrics characterize the celerity or the effect of an event on streamflow, not the velocity of water movement through the catchment (McDonnell and Beven, 2014). Because the runoff coefficient $Q/P$ and the ratio of quickflow to precipitation can

be calculated from hydrometric data alone, they have been estimated for many events and sites. Runoff coefficients $Q/P$ typically increase with storm size (e.g., Fischer et al., 2017; Jordan, 1994) and antecedent wetness (e.g., Litt et al., 2015; Sidle et al., 1995), and have been found to be unaffected by catchment area (e.g., McGlynn et al., 2004). The ratio of quickflow to precipitation has often been found to increase with storm size and intensity (e.g., Blume et al., 2007; Norbiato et al., 2009), with wetter antecedent conditions (e.g., Detty and McGuire, 2010; Merz et al., 2006; Penna et al., 2011;

von Freyberg et al., 2014) and with catchment area (e.g., Brown et al., 1999). However, it remains unclear whether these relationships arise because certain storm characteristics, antecedent wetness conditions, landscape properties, etc., facilitate more efficient transmission of recent precipitation ("event water") to the stream, or more effective mobilization of pre-event water from catchment storage. This question cannot be answered with hydrometric data alone; instead it also requires using tracer data to track the flow of water through the catchment, and thus to separate the runoff coefficient into its event and pre-

event components.





Tracer-based hydrograph separation uses stable water isotopes ($^2$H, $^{18}$O) to estimate the relative contributions of recent precipitation (event water, $Q_e/Q$) and catchment storage (pre-event water, $Q_{pe}/Q$) to streamflow (e.g., Klaus and McDonnell, 2013). Particularly in humid environments, stable water isotopes are considered to be nearly conservative tracers, because isotopic fractionation effects due to evaporation are negligible, so mixing is the major process controlling the isotopic

composition of streamwaters. Hydrograph separation studies have related the event-water fraction of discharge ($Q_e/Q$) to storm characteristics (e.g., James and Roulet, 2009; Pellerin et al., 2008), antecedent catchment wetness conditions (e.g., Casper et al., 2003; James and Roulet, 2009; Penna et al., 2015; Shanley et al., 2002; von Freyberg et al., 2017), landscape properties (e.g., Buttle, 1994; Fischer et al., 2017), and catchment size (e.g., Laudon et al., 2007; Shanley et al., 2002). Antecedent wetness has frequently been identified as a major control on the relative contribution of event water to

streamflow. Wetter antecedent conditions typically lead to smaller event water fractions $Q_e/Q$ and, by implication, larger pre-event water fractions. This suggests that under these conditions, larger volumes of pre-event water are available in more permeable subsurface layers that can be rapidly activated by incoming precipitation (e.g., Casper et al., 2003; Klaus and McDonnell, 2013; Muñoz-Villers and McDonnell, 2012). In contrast, at sites where positive relationships between $Q_e/Q$ and antecedent wetness have been observed, it has been hypothesized that vertical infiltration must first replenish storage deficits

before event water can be rapidly transported via lateral flow pathways or surface runoff (Shanley et al., 2002; von Freyberg et al., 2017). Still other studies have found no correlation between antecedent wetness and $Q_e/Q$ (Ocampo et al., 2006), which has often been attributed to an insufficient number of sampled events or an insufficient range of antecedent moistures (e.g., Barthold et al., 2017; Fischer et al., 2017; James and Roulet, 2009; Penna et al., 2015).

Although the runoff coefficient $Q/P$ and the event-water fraction $Q_e/Q$ have commonly been used in catchment studies, the

ratio between $Q_e$ and $P$ might provide an alternative tool for studying streamflow responses to precipitation events. The conceptual differences between the ratios $Q_e/Q$ and $Q_e/P$ can be explained by analogy with forward and backward transit times (e.g., Hrachowitz et al., 2016). A streamwater parcel's backward transit time is its age since entry as precipitation at the time it exits the catchment, whereas a precipitation parcel's forward transit time is the age it will accumulate in the catchment before it eventually exits as streamflow. Analogously, $Q_e/Q$ quantifies the fraction of total storm discharge that

comes from recent precipitation ("event water") – that is, the fraction of streamflow with a short backward transit time. The ratio $Q_e/P$, on the other hand, quantifies the fraction of total storm precipitation that will be discharged during the same event – that is, the fraction of precipitation with a short forward transit time.

The two quantities $Q_e/Q$ and $Q_e/P$ are not the same, for the simple reason that not all precipitation is discharged during the event (otherwise there would be no recharge of storage that supplies baseflow between events). Thus at the event scale, total

$P$ is typically greater than total $Q$, so $Q_e/P$ will be less than $Q_e/Q$. To use a concrete example: a 60 mm storm event might produce a streamflow response of 24 mm, from which 6 mm is event water ($Q_e$) from precipitation and the remaining 18 mm





of discharge is pre-event water ($Q_{pe}$) mobilized from catchment storage; the 54 mm of precipitation that does not form event discharge then goes into storage. In this example, the "backward" event water fraction $Q_e/Q$ would be 6 mm / 24 mm = 0.25 and the "forward" event water fraction $Q_e/P$ would be 6 mm / 60 mm = 0.1.

In addition to the ratio $Q_e/P$, we can also calculate the ratio of pre-event water volume to precipitation ($Q_{pe}/P$). Whereas the

event and pre-event water fractions of discharge are mathematically linked through the relationship $Q_e/Q+Q_{pe}/Q=1$, the ratios $Q_e/P$ and $Q_{pe}/P$ instead sum to the runoff coefficient $Q/P$. Care has to be taken with the conceptual definition of $Q_{pe}/P$: it quantifies how much pre-event water is mobilized by, not contained in, a unit volume of precipitation (which by definition cannot contain pre-event water). This highlights an important distinction between $Q_{pe}/P$ and the other three ratios ($Q_e/Q$, $Q_{pe}/Q$ and $Q_e/P$). In the event and pre-event water fractions of discharge ($Q_e/Q$ and $Q_{pe}/Q$), as well as in the event water

fraction of precipitation ($Q_e/P$), the numerator is physically (not just mathematically) a fraction of the denominator. That is, $Q_e$ is physically derived from storm precipitation, and along with $Q_{pe}$ is physically part of storm discharge. But $Q_{pe}$ is not physically part of $P$, and thus $Q_{pe}/P$ does not represent a physical fraction of a whole.

Although $Q_e/P$ has been estimated for multiple events at some catchments (Buttle and Peters, 1997; Eshleman et al., 1993; Rodhe, 1987; Pellerin et al., 2008), only Ocampo et al. (2006) have presented values of $Q_{pe}/P$ across several storms and

compared the variability in $Q_{pe}/P$ to antecedent wetness conditions. It remains to be tested whether the ratios $Q_e/P$ and $Q_{pe}/P$ are more informative metrics for studying catchment storm responses, compared to the widely used event water fraction of discharge $Q_e/Q$ or the runoff coefficient $Q/P$. Thus, in this paper, we compare runoff coefficients $Q/P$ and the different fractions $Q_e/Q$, $Q_{pe}/Q$, $Q_e/P$ and $Q_{pe}/P$ across 24 storm events and analyze their relationship with storm characteristics and initial catchment state. Based on these relationships we discuss possible streamflow generation processes at our study site

and highlight potential benefits of using $Q_e/P$ and $Q_{pe}/P$ over $Q/P$ and $Q_e/Q$. Our source data consist of high-frequency isotope measurements from a steep, partly forested headwater catchment in central Switzerland, collected during snow-free periods between September 2016 and October 2017.

## 2 Methods

### 2.1 Field site

The Erlenbach research catchment is located in the northern Swiss pre-Alps roughly 40 km south of Zurich. The 0.7 km² catchment spans an elevation range from 1100 to 1655 m a.s.l (Figure 1). The bedrock geology consists of alternating layers of conglomerates, clay and marl shales, limestone and calcareous sandstones, with lithological boundaries nearly perpendicular to the main valley axis (Hantke, 1967; Hsü and Briegel, 1991). Due to the layering and the diversity of the bedrock material, the landscape in the upper half of the catchment is divided into a sequence of steep hillslopes and plateaus.





On the plateaus, groundwater tables are shallow and mollic Gleysols with a carbonate-rich topsoil predominate; the hillslopes are mostly covered with macropore-rich umbric Gleysols (Schleppi et al., 1998). Overall, the soils reach maximum depths of around 2.5 m in depressions on the plateaus, while soil depths are shallower (0.5-1 m) on steeper slopes (Fischer et al., 2015; Rinderer et al., 2017). Due to the high clay content of the flysch material, the bedrock and soils are

generally characterized by low permeabilities and are prone to waterlogging. As a consequence, soil saturation is likely to occur on approximately 30% of the catchment area (Swiss Federal Office for the Environment (FOEN), 2011). The distribution of these areas agrees well with the mapped locations of numerous wet meadows that cover roughly 22 % of the catchment area (Figure 1a). The channel network in the Erlenbach catchment is dense (around 20 km/km$^2$), partly due to open drainage ditches in meadows as well as numerous small springs emerging from the bases of hillslopes in the upper part

of the catchment. The Erlenbach catchment is mainly covered with coniferous forests (53%) on the hillslopes, while partially forested areas (22%) and meadows (25%) occur largely on the plateaus (Fischer et al., 2015; Figure 1b).

Annual precipitation at Erlenbach averages roughly 1850 mm yr$^{-1}$ (period 2000-2015), with monthly maxima and minima in June and October, respectively (van Meerveld et al., 2018). In years with cold winters, up to 40 % of annual precipitation falls as snow (Stähli and Gustafsson, 2006), and in summer, frequent thunderstorms generate pronounced peaks in

streamflow.

## 2.2 Data set

A meteorological station is located at 1216 m a.s.l. on a grassland site near the south-western catchment border (Figure 1). At this station, precipitation is measured at 1.5 m above ground with a heated tipping bucket (Joss-Tognini 15183; Lambrecht meteo GmbH, Göttingen, Germany) and groundwater table depth is measured in a fully-screened pipe with a

pressure transducer (BTL2-P1-1000-B-S50; Balluff GmbH, Neuhausen a.d.F., Germany). The pressure transducer readings have not been calibrated against manual measurements, and thus should be considered only as a relative indication of groundwater table variations. Discharge has been monitored at the catchment outlet with a concrete flume since 1984 (Hegg et al., 2006). All measurements are recorded at 10-minute intervals and were aggregated to 30-minute or 1-hour intervals in the following analysis.

Soil moisture content is measured every 1 minute at two locations in the Erlenbach catchment, a grassland site (at the meteorological station, 1216 m a.s.l.) and a forest site (1185 m a.s.l.; Figure 1). At each site, one Decagon 10HS and three Decagon 5TM probes were installed at roughly 50 cm depth across an area of 3x4 m. Soil moisture data collection at the forested plot began only in December 2016. Prior to installation, the factory-calibrated soil moisture probes were compared to each other by installing them together in a bucket filled with moist sand. Only those probes that showed similar values

were installed in the field. As we were only interested in the temporal variations of soil moisture content over the course of



individual events, we considered the factory calibration of the soil moisture probes sufficient for our purposes. The soil moisture measurements from the four probes at each site were averaged to smooth out anomalous variations in soil moisture at the individual sampling locations, and subsequently aggregated to 30-minute or 1-hour intervals.

Recent technological advances allow for on-site high-frequency sampling and analysis of stable water isotopes in
streamwater and precipitation (von Freyberg et al., 2017). With such a lab in the field, isotopic responses in streamflow can be captured over long periods spanning a wide range of hydro-climatic conditions. A field laboratory was installed just above the Erlenbach outlet to measure stable water isotopes at 30-minute intervals. Streamwater was continuously pumped from the stream to the field laboratory. Precipitation was sampled with a 45-cm diameter funnel, flowed into the field lab through a tygon tube, and collected in a 1L teflon-lined separating funnel. Whenever the sample volume exceeded 50 mL,
the field lab alternately analysed precipitation and streamwater (thus yielding one precipitation and one streamwater sample every hour). During rainless periods streamwater was analysed on a 30-minute cycle; a drift correction check-standard was analysed every four hours.

The setup of the field laboratory is similar to the system described in von Freyberg et al. (2017) and will therefore be described here only briefly. A new analysis cycle began every 30 minutes by an automated pumping routine that filled a
50 mL-collection vessel with either streamwater or precipitation. A suction tube with a PE-filter frit (20 μm pore size) led from the collection vessel to two programmable high-precision dosing pumps (800 Dosino, hereafter simply "Dosino"; Metrohm AG, Herisau, Switzerland). These alternatingly took up 30 mL from the collection vessel and injected it at a constant flow rate of 1 mL min$^{-1}$ into a continuous water sampler module (CWS; Picarro Inc., Santa Clara, CA, USA) connected to a wavelength-scanned cavity ring-down spectrometer (CRDS; Picarro model L2130-i). During the 30-minute
injection cycle, the 50-mL collection vessel, the other Dosino, and all tubing were flushed with nanopure water and air, and then rinsed with approximately 10 mL of the next sample, to minimize sample carryover effects.

Within the CWS, liquid samples flow through a semipermeable membrane tube that is surrounded by a flow of dry air. Thus, water vapor diffuses steadily through the membrane wall and is transported with the dry-air flow to the isotope analyser. The CRDS measures concentrations of $^{18}$O and $^2$H every 6 s, however, we averaged the last 10 minutes of each 30-
minute injection period to obtain the final isotope values. The measurement precision of the CWS coupled to the CRDS with the Dosino injection system was estimated from the standard deviations of these 10-minute averages (von Freyberg et al., 2017). For $\delta^{18}$O and $\delta^2$H, the averages of these standard deviations were 0.09 ‰ and 0.21 ‰, respectively, which are used as estimates of uncertainty in the error propagation calculations (Sect. 2.4). Isotopic abundances are reported in $\delta^{18}$O and $\delta^2$H relative to Vienna standard mean ocean water (VSMOW). The isotope analyzer was initially calibrated to
VSMOW-SLAP standards before installation in the field laboratory; during the field deployment, instrument drift and carry-



over were quantified and corrected by regularly measuring one internal isotope standard every four hours and two internal standards approximately weekly.

## 2.3 Event properties: storm characteristics and initial catchment state

The following metrics were used to describe the storm characteristics: total event precipitation ($P$, mm), cumulative precipitation before peak flow ($P_{untilQpeak}$, mm), mean precipitation intensity ($P_{int}$, mm/hr), maximum precipitation over 1 hour ($P_{1h}$, mm hr$^{-1}$) and 4 hours ($P_{4h}$, mm), and event duration ($T$, hours). The antecedent catchment wetness state was quantified using three-day and seven-day antecedent precipitation ($AP3$ and $AP7$, mm), as well as the 1-hour average values of discharge ($Q_{ini}$, mm hr$^{-1}$), groundwater table depth ($GW_{ini}$, cm), and soil moisture ($SM_{ini}$, percent) before the onset of the storm event. The end of a storm event was defined as the time that (i) event water discharge declined to 5 % of its peak value or (ii) another precipitation event began, whichever came first; case (i) prevailed for 18 of 24 events.

We compared these metrics of storm characteristics and antecedent wetness conditions with the event and pre-event water fractions using Spearman rank correlation. Spearman rank correlation was preferred over Pearson (product-moment) correlation because it is less sensitive to extreme values and thus more suitable for smaller data sets. For our analyses, correlation $p$-values of $p<0.01$ were considered statistically significant.

## 2.4 Two-component hydrograph separation to quantify event- and pre-event water volumes

We use the concentrations of the stable water isotopes $^2$H and $^{18}$O in streamwater and precipitation to separate the streamflow hydrograph into two components, event water and pre-event water. Event water (subscript e) is streamwater that entered the catchment as precipitation during a storm event, whereas pre-event water (subscript pe) is streamwater derived from catchment storage. Following Pinder and Jones (1969), tracer-based hydrograph separation is based on a mass balance for water,

$$q_i = q_{e_i} + q_{pe_i} \qquad , \tag{1}$$

and tracer flux,

$$c_i q_i = c_{e_i}^* q_{e_i} + c_{pe_i} q_{pe_i} \quad , \tag{2}$$

Where $q_i$, $q_{e_i}$, and $q_{pe_i}$ are total, event, and pre-event water fluxes at time step $i$, and $c_i$, $c_{e_i}^*$, and $c_{pe_i}$ are the tracer concentrations in total streamflow, event water and pre-event water, respectively.

The fraction of event water in discharge for each time step $i$ can thus be obtained by combining Eqs. (1) and (2):





$$\frac{q_{e_i}}{q_i} = \frac{c_i - c_{pe_i}}{c_{e_i}^* - c_{pe_i}} \quad . \tag{3}$$

To make the $\frac{q_{e_i}}{q_i}$ time series continuous, it is linearly interpolated between measurement gaps that occur whenever check standards or precipitation samples are analysed instead of streamwater samples.

Following general practice, we assume that the tracer concentration of pre-event water is constant over the duration of the

5 event (Sklash and Farvolden, 1979), so that $c_{pe_i}$ becomes $c_{pe}$. We calculate $c_{pe}$ as the average isotopic composition of the last five streamwater samples before the beginning of each storm event. We estimate the time series of event water tracer concentration, $c_{e_i}^*$ , as an incremental weighted mean (McDonnell et al., 1990), weighted by precipitation rates over all previous time steps $j$ since the beginning of the storm:

$$c_{e_i}^* = \frac{\sum_{j=k}^{i} P_j c_{e_j}}{\sum_{j=k}^{i} P_j} \quad , \tag{4}$$

10 The total event water volume ($Q_e$) is the cumulative sum of the individual values $q_{e_i}$ over the event duration, and the cumulative pre-event water volume ($Q_{pe}$) was obtained by subtracting $Q_e$ from the total discharge volume $Q$ over the event.

The ratios of event and pre-event water relative to precipitation are

$$\frac{Q_e}{P} = \frac{Q_e}{Q} \cdot \frac{Q}{P} \tag{5}$$

and

$$\frac{Q_{pe}}{P} = \left(1 - \frac{Q_e}{Q}\right) \cdot \frac{Q}{P} \quad . \tag{6}$$

The standard errors ($SE$) of $c_{e_i}^*$ and $\frac{q_{e_i}}{q_i}$ were estimated though Gaussian error propagation (Genereux, 1998):

$$SE(c_{e_i}^*) = \left[\frac{\sum_{j=k}^{i} P_j \left(c_{e_j} - c_{e_j}^*\right)^2}{(j-k)\sum_{j=k}^{i} P_j}\right]^{\frac{1}{2}} \quad , \tag{7}$$



and

$$SE(\frac{q_{e_i}}{q_i}) = \left\{ \left[ \frac{-1}{c_{pe} - c^*_{e_i}} SE(c_i) \right]^2 + \left[ \frac{c_i - c^*_{e_i}}{(c_{pe} - c^*_{e_i})^2} SE(c_{pe}) \right]^2 + \left[ \frac{c_p - c_i}{(c_{pe} - c^*_{e_i})^2} SE(c^*_{e_i}) \right]^2 \right\}^{\frac{1}{2}} . \tag{8}$$

The standard error of $c_{pe}$ is estimated by pooling the uncertainty in the individual measurements, and their standard deviation from one another (von Freyberg et al., 2017). Because $c_i$ and $c_{pe}$ are independent measurements, their errors $SE_{c_i}$

5 and $SE_{c_{pe}}$ are likely to be uncorrelated with each other. In contrast, errors in the calculated event water isotope values $c^*_{e_i}$ will be highly correlated with each other over time due to the incremental volume-weighting of tracer concentrations in precipitation. Taking these correlations into account requires first-order, second moment error propagation (Bevington and Robinson, 2003), which reduces to Gaussian error propagation in the special case of uncorrelated errors. The first-order, second moment error propagation formula for the event water fraction $Q_e/Q$, averaged over all times $i$ in the storm event

10 (Sect. 2.3), is

$$SE\left(\frac{Q_e}{Q}\right) = \left\{ \sum_i \left[ \frac{q_j}{Q} \cdot \frac{SE(c_i)}{c^*_{e_i} - c_{pe}} \right]^2 + \sum_i \left[ \frac{q_i}{Q} \cdot \frac{c_i - c^*_{e_i}}{(c^*_{e_i} - c_{pe})^2} \cdot SE(c_{pe}) \right]^2 + \left[ \sum_i \left( \frac{q_i}{Q} \cdot \frac{c_{pe} - c_i}{(c^*_{e_i} - c_{pe})^2} \cdot SE(c^*_{e_i}) \right) \right]^2 \right\}^{\frac{1}{2}} , \tag{9}$$

where the square brackets on the outside of the last summation, rather than the inside, reflect the conservative assumption that the errors in the event water isotope values $c^*_{e_i}$ are perfectly correlated. The other two pairs of square brackets are inside the summations, reflecting the assumption that the errors in $c_i$ and $c_{pe}$ are uncorrelated. For simplicity, and because our

15 main focus is on the event and pre-event fractions of the water fluxes rather than the fluxes themselves, we ignore any measurement errors in $q_i$ and $P$.

Following Eqs. (5) and (6), the standard errors of $Q_e/P$ and $Q_{pe}/P$ can be estimated with:

$$SE\left(\frac{Q_e}{P}\right) = SE\left(\frac{Q_e}{Q}\right) \cdot \frac{Q}{P}, \tag{10}$$

and

$$SE\left(\frac{Q_{pe}}{P}\right) = SE\left(\frac{Q_{pe}}{Q}\right) \cdot \frac{Q}{P} . \tag{11}$$



## 3   Results and Discussion

Figure 2 shows the time series of the observed variables for the roughly 8-month study period 15Sep2016 to 01Nov2017, excluding the winter season influenced by snowfall and snow cover (06Nov2016 – 05May2017). River discharge responds promptly to precipitation and is strongly synchronized with shallow soil moisture and groundwater table variations. The

average soil moisture content at the grassland site was slightly higher and more variable than at the forested site. The values of stable water isotopes in precipitation are highly variable within and across events, ranging between -170.3 and -31.7 ‰ for $\delta^2H$, and between -22.5 and -4.2 ‰ for $\delta^{18}O$ for the storms considered here. Streamwater isotopes are strongly damped, but also show distinct responses to individual storms (Figure 2a). For the 24 events, values of $\delta^2H$ and $\delta^{18}O$ in streamwater ranged from -56.1 to -84.3 ‰ and from -8.6 to -12.2 ‰, respectively. During the roughly 8-month study period, more than

9,400 water samples were measured; missing values, due to problems with the automatic sampling cycle or instrument malfunctioning, account for roughly 8 % of the data set.

### 3.1 General properties of the events

Table 1 provides an overview of the storm characteristics and antecedent wetness conditions for the individual storm events. Total storm rainfall $P$ ranged between 8.2 and 63.2 mm ($25.1 \pm 3.1$ mm, mean±$SE$) and total discharge $Q$ ranged between 0.5

and 25.6 mm ($9.8 \pm 1.7$). During the individual storm events, the 4-hour peak discharge reached values between 0.11 and 12.5 mm. Figure 3 shows that the 24 storm events used for our analysis span a wide range of flow regimes.

The various metrics of catchment antecedent wetness conditions were highly correlated with each other. Spearman rank correlation coefficients were $\rho > 0.60$ ($p < 0.002$) for all metrics and combinations except for $AP7$ and $GW_{ini}$ ($\rho = 0.49$, $p = 0.015$). Initial soil moisture $SM_{ini}$, initial groundwater levels $GW_{ini}$, and seven-day antecedent precipitation ($AP7$) correlate strongly

($\rho > 0.83$ in all cases) with initial discharge ($Q_{ini}$), suggesting that these measures of antecedent wetness are representative, at least as relative indicators, at the catchment scale.

### Both isotopes yield similar hydrograph separation results

Figure 4a shows that $\delta^{18}O$ and $\delta^2H$ yield instantaneous event water fractions of discharge ($\frac{q_{e_i}}{q_i}$) at peak flow that are equal within error for each event, except for 22Oct2017, for which the difference is 21 % greater than twice their pooled

uncertainties. For $Q_e/Q$ we also observe a good agreement between both isotopes, except for the events on 25Jun2017, 25Sep2017, 26Oct2017 and 29Oct2017 for which the differences are 178 %, 4 %, 2 % and 2 % greater than twice their pooled uncertainties, respectively (Figure 4b, Table S1 in the Supplement). We thus assume that inferences derived from the two isotopes will be consistent with each other. Measurements of $\delta^2H$ were less noisy than those of $\delta^{18}O$ relative to their



respective ranges of variability, so values such as $\frac{q_{e_i}}{q_i}$ and $Q_e/Q$ will be more precise when derived from $\delta^2$H rather than $\delta^{18}$O.
Therefore, the following analysis is performed based on $\delta^2$H; $\delta^{18}$O would yield similar results but with larger uncertainties.

**Hydrograph separation results for 24 storm events**

Figure 5 and Table 2 compare the storm events' runoff coefficient $Q/P$ and show that total storm discharge is typically less
than half of total storm precipitation, and in some cases much less. On average, runoff coefficients are 0.34±0.04
(mean±$SE$), but their storm-to-storm variability is large (0.03 to 0.72), suggesting that the effectiveness with which
precipitation signals are converted to discharge responses varies considerably at Erlenbach.

The relative fractions of event water in discharge ($Q_e/Q$) are highly variable across the 24 storm events, ranging from 0.04 to
0.75, with a mean value of 0.23±0.04. The relative contribution of event water to discharge exceeded 50% for only two
storms (Figure 5), and on average, discharge at Erlenbach was comprised of roughly 77% pre-event water. Similarly high
pre-event water fractions relative to discharge have been observed at other humid forested headwater catchments (e.g.,
Brown et al., 1999; Buttle, 1994; Jones et al., 2006; McGlynn and McDonnell, 2003).

For all 24 storms, the event water fractions of precipitation $Q_e/P$ are smaller than the corresponding event water fractions of
discharge, for the simple reason that $P$ exceeds $Q$ (Table 2). The values of $Q_e/P$ range from 0.002 to 0.34 (mean±$SE$
0.08±0.02) while the pre-event water volume relative to precipitation ($Q_{pe}/P$) ranges from 0.03 to 0.68 (mean±$SE$ 0.28±0.03).
This suggests that on average, each precipitation event at Erlenbach activated pre-event water equal to roughly a third of the
rainfall volume, while the event water contribution to streamflow accounted for less than 10 % of the rainfall volume. Thus,
precipitation had a nearly three-fold larger effect on the activation of pre-event water than on the transmission of event water
to the stream.

Relatively few stable isotope studies have analyzed numerous events at high temporal resolution (e.g., Birkel et al., 2012;
Fischer et al., 2017; Ocampo et al., 2006; von Freyberg et al., 2017), revealing large variations in the relative amounts of
event and pre-event water from storm to storm. At Erlenbach, we find that the event water fraction of discharge $Q_e/Q$ is
much more variable, relative to its mean (coefficient of variation $CV$=0.74), than the pre-event water fraction $Q_{pe}/Q$
($CV$=0.23). This follows as a direct consequence of $Q_e/Q$ being smaller, on average, than $Q_{pe}/Q$, and from these two
quantities being complements of one another ($Q_{pe}/Q$=1- $Q_e/Q$), implying that their standard deviations must be equal. Event-
and pre-event water volumes relative to precipitation are more variable across storms, ($Q_e/P$ $CV$=0.96 and $Q_{pe}/P$ $CV$=0.61),
suggesting that the "forward" event- and pre-event water fractions ($Q_e/P$, $Q_{pe}/P$) might be more informative, for instance
when used for correlation analyses, compared to the less variable "backward" event- and pre-event water fractions of





discharge ($Q_e/Q$, $Q_{pe}/Q$). More fundamentally, $Q_{pe}/Q$ and $Q_e/Q$ contain completely redundant information, because they are linear functions of one another. By contrast, $Q_e/P$ and $Q_{pe}/P$ do not sum to a constant (instead they sum to the runoff coefficient), so they each contain distinct information.

**Detailed description of three contrasting events**

To investigate the conceptual differences of the ratios $Q_{pe}/Q$, $Q_e/P$ and $Q_{pe}/P$ in more detail, Figure 6 shows the hydrograph separation results for three storm events, 02Oct2016, 05Oct2017, and 10Jul2017, along with the time series of precipitation, river discharge, soil moisture, and $\delta^2H$ values in precipitation and streamwater. During the 02Oct2016 storm, antecedent moisture conditions were dry ($AP7$=11 mm) and total precipitation ($P$) and river discharge ($Q$) were 21.6 mm and 4.8 mm, respectively, resulting in a runoff coefficient $Q/P$ of 0.22 (Figure 6a). During the 05Oct2017 storm, antecedent conditions

were wetter ($AP7$=69 mm), and consequently 33.5 mm of rain produced 20.5 mm of discharge, yielding a runoff coefficient of 0.61; roughly 50 % more rain generated roughly 300 % more discharge, relative to the earlier event (Figure 6b). The response times of river discharge to incoming rainfall, measured here as the time it takes for $q_i$ to increase by more than 30 % relative to $Q_{ini}$, were similar for both storm events (2 vs. 2.5 hours), as were the changes in soil moisture recorded at the grassland site. The instantaneous event water fractions of discharge $\dfrac{q_{e_i}}{q_i}$ peaked at similar values in the two events

(0.30±0.01 and 0.33±0.01, respectively), and the aggregated event water volumes relative to discharge ($Q_e/Q$) were likewise similar (0.23±0.01 and 0.24±0.01). Thus, river discharge was predominantly pre-event water during both events, and despite the great differences in total event rainfall and antecedent wetness conditions, both storms resulted in similar event water fractions of discharge (Table 2). In contrast, the event and pre-event water volumes relative to precipitation were roughly three times higher in the second storm ($Q_e/P$=0.15 vs. 0.05, and $Q_{pe}/P$=0.46 vs 0.17; Table 2), suggesting that $Q_e/P$ and $Q_{pe}/P$

might more clearly reflect how catchments respond to variations in antecedent wetness conditions and total event rainfall.

During the 10Jul2017 storm, antecedent conditions were slightly wetter ($AP7$=20.2 mm) than on 02Oct2016, whereas the total rainfall volume and the runoff coefficient were intermediate to those of the two October events ($P$=25.4 mm, $Q/P$=0.28). However, because the maximum 1-hour rainfall intensity during the 10Jul2017 storm was nearly four times larger compared to the two October events, peak flow rates during the 10Jul2017 storm were similar to the much larger

05Oct2017 storm (Figure 6b, c). In contrast to both the 02Oct2016 and 05Oct2017 storms, event water comprised nearly 50 % of discharge during the 10Jul2017 storm. Similarly, the ratios of event and pre-event water relative to precipitation ($Q_e/P$=0.135 and $Q_{pe}/P$=0.141, respectively) indicate that the 10Jul2017 storm event mobilized equivalent volumes of event and pre-event water. This suggests that infiltration excess during the high-intensity precipitation period enhanced the direct contribution of event water to the stream.


Across all 24 events, there is a general tendency for the instantaneous event water fraction $\frac{q_{e_i}}{q_i}$ to peak on the rising limb, ahead of the flow peak (Figs. S1-S4 in the Supplement; for the three storms 02Oct2016, 05Oct2017, and 10Jul2017 the peak times are indicated by grey and black vertical arrows in Figure 6a-f). Thus the event water fraction at the time of peak flow was typically somewhat smaller than the peak event water fraction. This observation shows the importance of evaluating

event water fractions over the entire hydrograph rather than just at peak flow (von Freyberg et al., 2017). It also suggests that peak flows are generated primarily by mobilizing pre-event water, which dilutes the event water that is more prominent on the rising limb of the hydrograph.

### 3.2 Catchment responses to storm characteristics and antecedent wetness

**Runoff coefficients $Q/P$ depend on antecedent wetness, not storm size**

To identify the main controls on the relative contribution of event- and pre-event water to catchment outflow, we analyse their correlations with storm characteristics and catchment antecedent wetness conditions (Figure 7, Table 3). Larger, longer, and more intense storms result in larger discharges $Q$, while there is no strong effect of antecedent precipitation ($AP3$, $AP7$), antecedent discharge ($Q_{ini}$), antecedent soil moisture ($SM_{ini}$), or antecedent groundwater table depth ($GW_{ini}$) on $Q$ (Table 3). In contrast, although the runoff coefficient $Q/P$ does not seem to be affected by storm size, it is strongly

positively correlated with all metrics of catchment antecedent wetness conditions. This indicates that wetter conditions enhance the efficiency with which precipitation inputs trigger increases in streamflow. Clarifying the mechanisms behind this phenomenon requires not just hydrometric measurements, but also tracers that track the water flow paths through the catchment.

**Event water contributions to streamflow are controlled by storm characteristics, not antecedent wetness**

Event water fractions of discharge and precipitation ($Q_e/Q$ and $Q_e/P$) at Erlenbach show statistically significant positive correlations with most storm characteristics, i.e. $P$, $P_{untilQpeak}$, $P_{1h}$ and $P_{4h}$ (Table 3). These relationships suggest that event water discharge increases more-than proportionally with storm size. Similar results have been reported for $Q_e/Q$ in forested and urban catchments (James and Roulet, 2009; Pellerin et al., 2008; Penna et al., 2015) and it has been hypothesized that more incoming rainfall eventually triggers saturation or infiltration excess, which leads to more surface runoff. Rainfall

intensity has also been reported to affect $Q_e/Q$ (Eshleman et al., 1993; Waddington et al., 1993), and at Erlenbach we find strong positive correlations of $Q_e/Q$ and $Q_e/P$ with one-hour and four-hour peak precipitation intensity ($P_{1h}$ and $P_{4h}$). We do not identify a strong relationship with the average rainfall intensity $P_{int}$, probably because its definition (total volume divided by total storm duration) makes it strongly dependent on the duration of low-intensity rainfall that contributes little to $Q_e$ or $Q$.





Perhaps surprisingly, the event water fraction of discharge is lower, not higher, under wetter antecedent conditions; correlations between $Q_e/Q$ and the antecedent wetness metrics range from -0.24 to -0.51 but are not, however, statistically not significant ($p>0.01$; Table 3). Indeed, even the volume of event water $Q_e$ by itself (not as a fraction of total $Q$) does not become systematically larger under wetter conditions; the correlations between $Q_e$ and the antecedent wetness metrics range

from 0.01 to 0.13 (Table 3). Thus, wetter antecedent conditions do not lead to systematically higher event water discharges in either absolute or relative terms. The negative correlation between antecedent wetness and $Q_e/Q$ arises for the simple reason that wetter antecedent conditions increase total discharge (primarily by mobilizing more pre-event water), while $Q_e$ remains largely unchanged.

**Pre-event contributions relative to discharge correlate weakly with antecedent wetness**

Because the pre-event water fraction of discharge ($Q_{pe}/Q$) is defined as the complement to the event water fraction ($Q_{pe}/Q=1-Q_e/Q$), its correlations with storm properties and antecedent wetness will be opposite to those of $Q_e/Q$. At Erlenbach, $Q_e/Q$ is weakly negatively correlated (and thus $Q_{pe}/Q$ is weakly positively correlated) with our metrics of antecedent wetness. The positive correlation between antecedent wetness and $Q_{pe}/Q$ suggests that a greater volume of pre-event water is available under wet conditions. However, the relationships between $Q_{pe}/Q$ or $Q_e/Q$ and antecedent wetness are

highly scattered, consistent with other studies (Fischer et al., 2017; James and Roulet, 2009; Ocampo et al., 2006). Thus, these relationships are not much help in explaining why runoff coefficients at Erlenbach are strongly correlated with antecedent wetness and not with storm size and intensity.

**Ratio of pre-event water to precipitation correlates strongly with antecedent wetness**

In contrast to the event- and pre-event water fractions of discharge ($Q_e/Q$, $Q_{pe}/Q$), the event water fraction of precipitation

($Q_e/P$) correlates strongly with metrics of storm characteristics (but not antecedent wetness), and the ratio of pre-event water to precipitation ($Q_{pe}/P$) correlates strongly with metrics of antecedent wetness conditions (but not storm characteristics; Figure 7, Table 3). We find positive relationships between $Q_e/P$ and most metrics of storm characteristics, such as $P$, $P_{until Qpeak}$, and $P_{4h}$, similar to the relationships found for $Q_e/Q$. We also find that $Q_{pe}/P$ – but not $Q_e/P$ – is strongly (positively) correlated with all of our metrics of antecedent wetness conditions ($Q_{ini}$, $GW_{ini}$, $SM_{ini}$, $AP3$ and $AP7$). The

correlations between $Q_{pe}/P$ and storm characteristics are much weaker, suggesting that the activation of pre-event water by precipitation is primarily controlled by pre-storm wetness conditions and not by storm size.

The runoff coefficient's sensitivity to antecedent wetness, and its insensitivity to storm characteristics, can be understood through the behavior of $Q_e/P$ and $Q_{pe}/P$, which sum to the runoff coefficient itself: $Q/P=Q_e/P+Q_{pe}/P$. Because $Q_{pe}/P$ is larger and more variable than $Q_e/P$ (with one exception, all values of $Q_e/P$ are less than 0.2, whereas $Q_{pe}/P$ spans a range roughly



three times as large), variations in the runoff coefficient $Q/P$ will be dominated by variations in $Q_{pe}/P$. Thus, because $Q_{pe}/P$ is sensitive to antecedent moisture, so is the runoff coefficient. For example, seven-day antecedent moisture $AP7$ is much more tightly correlated with $Q_{pe}/P$ ($\rho=0.79$) than with $Q_e/P$ ($\rho=0.03$), and because the variability of $Q_{pe}/P$ is much greater than that of $Q_e/P$, it dominates the correlation between $AP7$ and the runoff coefficient $Q/P$ ($\rho=0.74$; Table 3). The same line

of argument explains why the runoff coefficient is relatively insensitive to storm size and intensity. For example, the correlation between 4-hour storm intensity $P_{4h}$ and $Q_e/P$ is 0.71, but the correlation with $Q_{pe}/P$ is -0.15, and because the range of variation in $Q_{pe}/P$ is roughly three times larger, the resulting correlation between $P_{4h}$ and the runoff coefficient $Q/P$ is only 0.09.

Storm characteristics and antecedent wetness generally exhibit stronger correlations with $Q_e/P$ and $Q_{pe}/P$ than with $Q_e/Q$ and

$Q_{pe}/Q$ (Figure 7, Table 3). For this reason, and because they are both components of the runoff coefficient itself, $Q_e/P$ and $Q_{pe}/P$ are more informative than $Q_e/Q$ or $Q_{pe}/Q$ in explaining how the runoff coefficient is controlled by event properties. One reason for the weaker correlations between storm characteristics and $Q_e/Q$ or $Q_{pe}/Q$ is that larger and more intense storms increase not only $Q_e$ and $Q_{pe}$, but also $Q$; thus the ratios $Q_e/Q$ and $Q_{pe}/Q$ will change less than one might expect from the effects on $Q_e$ and $Q_{pe}$ themselves. This points to an important limitation when using $Q_e/Q$ or $Q_{pe}/Q$ as descriptors of

catchment response during storm events and might explain why the relationships between $Q_e/Q$ or $Q_{pe}/Q$ and metrics of event properties often do not yield clear inferences about controlling factors for streamflow generation under different boundary conditions.

Our results provide important context for previous studies that have used the ratio between graphically estimated quickflow and precipitation as a proxy for how efficiently precipitation is translated into streamflow (e.g., Detty and McGuire, 2010;

James and Roulet, 2007; Merz et al., 2006; Norbiato et al., 2009; Penna et al., 2011; Taylor and Pearce, 1982; Sidle et al., 1995). Many of these studies have shown that the ratio between quickflow and precipitation increases with antecedent wetness, suggesting that wetter conditions lead to the activation of rapid flow pathways, as groundwater levels rise to shallower, more permeable layers, or as hydrologic connectivity increases due to expansion of the river network and connection of wetlands and hillslopes to the stream (e.g., Dunne and Black, 1970; Godsey and Kirchner, 2014; McGuire and

McDonnell, 2010). It should be stressed, however, that separating the hydrograph into baseflow and quickflow with graphical or digital filter methods is a highly subjective process (Blume et al., 2007) that leaves unresolved the critical question of whether this quick flow is primarily event water or pre-event water. Simply put: wetter conditions lead to more quickflow, but is this because more rainfall reaches the stream, or because more pre-event water is mobilized from catchment storage? Our results show that even at the highly dynamic Erlenbach catchment, antecedent wetness correlates

with $Q_{pe}/P$ but not $Q_e/P$; that is, wetter conditions lead primarily to more efficient mobilization of pre-event water, rather than more efficient transmission of rainfall to the stream.




### 3.3 Controls on the event and pre-event water at Erlenbach

Previous research in the Erlenbach catchment has identified two possible sources of base flow: springs in the uppermost part of the catchment and groundwater discharge from a shallow aquifer on top of the low-permeability bedrock (van Meerveld et al., 2018). However, conceptual models of streamflow generation at Erlenbach have not considered antecedent wetness

conditions as a major control on the discharge of pre-event water, possibly because tracer-based estimates of $Q_{pe}/Q$ did not correlate with various metrics of antecedent wetness (Fischer et al., 2017 and results presented here). By contrast, when $Q_{pe}/P$ is considered instead of $Q_{pe}/Q$, our data clearly show that antecedent wetness strongly controls the mobilization of pre-event water. Because pre-event water comprises a large fraction of streamflow, even during events, antecedent wetness conditions are thus an important control on the streamflow regime at Erlenbach (along with storm size and intensity).

The several-fold increase of $Q_{pe}/P$ with antecedent wetness implies that pre-event water is more efficiently mobilized under wetter conditions. The rapid activation of this stored pre-event water at Erlenbach (even during small storms) can be explained with generally shallow perched groundwater tables in the aquifer overlying the low-permeability bedrock. In a neighboring catchment, groundwater tables are usually less than 0.4 m below the ground surface in low-permeability soils on plateaus and at the bottoms of the hillslopes, and not much deeper in the more permeable hillslope soils (Rinderer et al.,

2014; Schleppi et al., 1998). Earlier studies at the Erlenbach and neighboring catchments showed that the low-permeability soils on the plateaus saturate first during events, and as a consequence, a mixture of event and pre-event water flows off as shallow subsurface stormflow and surface runoff (Rinderer et al., 2016; van Meerveld et al., 2018).

Our isotope-based hydrograph separation results show that the relative contribution of event water to discharge plays only a minor role in the streamflow regime of the Erlenbach, despite the runoff coefficients $Q/P$ being > 0.4 for more than half of

all storm events. Only for two storms with dry antecedent conditions and high-intensity rainfall did the event-water fraction of discharge $Q_e/Q$ exceed 0.5. While wetter antecedent conditions clearly facilitated the mobilization of more pre-event water, they did not significantly enhance bypass flow of event water via surface runoff on saturated areas.

### Source areas of event water

Several studies have used the ratio $Q_e/P$ as a proxy for the relative catchment area that generates event water (e.g., Buttle and

Peters, 1997; Ocampo et al., 2006), or have shown that $Q_e/P$ predicts the mapped extent of saturated or impermeable areas (e.g., Eshleman et al., 1993; Rodhe, 1987; Pellerin et al., 2008). Direct runoff of event rainfall can occur on impermeable and low-permeability surfaces, on saturated areas, and through preferential flow pathways (Beven and Germann, 1982; Dunne and Black, 1970). At Erlenbach, the channel network itself and an asphalt road account for roughly 1.2 % of the total catchment area; this is a plausible lower bound for the area that can generate surface runoff. In addition, surface runoff may



also occur on the saturated low-permeability soils of the plateaus, as well as in locations where the water table is close to the surface (depressions, bottoms of hillslopes, river banks; Rinderer et al., 2014). Mapped wet meadows occur on approximately 22 % of the catchment area, and geostatistical analyses suggest that around 30 % of the total catchment area is prone to saturation (Swiss Federal Office for the Environment (FOEN), 2011; Figure 1a); 30 % is thus a plausible upper

bound for the area that can generate surface runoff. The range between these upper and lower bounds is spanned by the 24 storms that we analysed, whose $Q_e/P$ varied between 0.002 and 0.34 (mean±$SE$ 0.08±0.02). This suggests that the variability of $Q_e/P$ across storms may reflect the contraction and expansion of these source areas and changes in their hydrological connectivity to the channel network.

**Precipitation primarily mobilizes pre-event water instead of running off to the stream**

For most of the analyzed storms, event water is a much smaller contributor to streamflow than pre-event water. This observation makes sense if we assume that most precipitation lands directly on the more permeable hillslope soils (which constitute most of the catchment), or reaches these soils by flowing down-gradient from low-permeability or saturated areas on the plateaus located above these hillslopes (Rinderer et al., 2014). Water infiltrating into the hillslope soils presumably raises the groundwater table into more permeable soil layers, facilitating rapid downslope transport of groundwater and

resulting in a mixture of event and pre-event water in the stream. However, during particularly large storms, such as those on 10Jul2017 and 18Aug2017, it is likely that event-water generating areas hydrologically connect to the channel network and event water becomes a much larger fraction of the streamflow hydrograph.

**Sources of pre-event water and link to antecedent wetness**

Temporal and spatial variations in subsurface hydrological connectivity have been studied in a 20 ha catchment adjacent to

Erlenbach, using a dense network of groundwater observation wells (Rinderer et al., submitted manuscript). That study showed that catchment areas with a subsurface hydrological connection to the channel network expand and contract during storm events. It seems likely that similar processes occur at Erlenbach as it shares similar landscape properties with Rinderer et al.'s study site. Following this line of thought, one can speculatively infer that infiltration of precipitation into the hillslopes and the mobilization of hillslope pre-event water significantly increase subsurface hydrologic connectivity. The

amount of hillslope pre-event water that is mobilized will therefore largely depend on the pre-storm storage deficit in the hillslopes, and not so much on the pre-storm storage deficit in the event-water source areas (where the storage deficit is always small). This would explain our observation that direct runoff of event water $Q_e$ to the stream is mainly controlled by storm characteristics (and not by antecedent wetness), whereas the mobilization of $Q_{pe}$ is strongly controlled by antecedent wetness (and much less by storm characteristics). However, no spatially distributed measurements of groundwater table

dynamics are available to further investigate this hypothesis at Erlenbach.



### 3.4 Fingerprints of catchment response

Our analysis, spanning 24 storm events with contrasting characteristics, reveals at least four patterns of behavior that could potentially be useful as "fingerprints" of catchment response if they are found to also hold in other catchments. First, the runoff coefficient $Q/P$ is a roughly linear function of the logarithm of antecedent discharge $Q_{ini}$ (Figure 7). If similar linear

relationships are also observed elsewhere, their slopes (which are dimensionless) could be used as indices of catchment response for catchment comparison purposes. Second, the event water fraction of precipitation $Q_e/P$ is a roughly linear function of $P$ itself (Figure 7), and its slope can be considered as an index of how storm size alters the fraction of the catchment area that is connected to the stream by fast flowpaths. One could even consider the hypothetical point where this linear relationship crosses the $Q_e/P = 1$ line (which would not be reached in practice) as an indicator of how much

precipitation would be required to establish fast flowpaths connecting the entire catchment to the stream. Third, the ratio of pre-event discharge to precipitation $Q_{pe}/P$ is a roughly linear function of antecedent precipitation $AP7$, and its slope could be considered as an index of how antecedent moisture alters the fraction of the catchment in which stored, pre-event water can be efficiently mobilized by streamflow. Fourth, the ratio of pre-event discharge to precipitation $Q_{pe}/P$ is a roughly linear function of the logarithm of antecedent discharge $Q_{ini}$ (Figure 7). The slope of this relationship, which is dimensionless,

contains information on how antecedent discharge reflects antecedent moisture, and how antecedent moisture determines the mobility of pre-event water. We emphasize that these "fingerprints" of catchment behavior are necessarily speculative, unless and until they are confirmed by cross-catchment comparisons. We also note that none of these four relationships can remain linear forever, because all of these ratios are logically constrained to be ≤1; thus they must become asymptotic at some point.

**Summary and outlook**

Tracer-based estimates of event and pre-event water fractions of discharge ($Q_e/Q$, $Q_{pe}/Q$) are often compared to catchment properties, storm characteristics, and antecedent wetness conditions to identify controls on streamflow generation mechanisms. However, these relationships may be obscured because the same factors that influence event discharge $Q_e$ and pre-event discharge $Q_{pe}$ also necessarily influence total discharge $Q$ as well. We thus propose that the fractions of event

water and pre-event water relative to precipitation ($Q_e/P$ and $Q_{pe}/P$) provide an alternative and more insightful approach to study catchment storm responses. Here, we use 30-minute stable water isotope data, collected over a period of roughly 8 months at the pre-Alpine Erlenbach catchment, to obtain robust estimates of $Q_e$ and $Q_{pe}$. In total, we analysed 24 rainfall events spanning a wide range of hydro-climatic conditions. Our findings are summarized below:





1. Pre-event water dominates streamflow for the majority of the storms. While $Q_e/Q$ strongly (positively) correlates with storm size and intensity, correlations between antecedent wetness conditions and $Q_e/Q$ are often weak and scattered. Because $Q$ is the sum of $Q_e$ and $Q_{pe}$, $Q$ itself is strongly affected by storm size and antecedent wetness conditions in similar ways as $Q_e$ and $Q_{pe}$. As a result, $Q_e/Q$ usually correlates only weakly with storm size and antecedent wetness,
hampering the identification of major controls on streamflow generation.

2. By relating event and pre-event water volumes to event precipitation $P$ instead (i.e., $Q_e/P$, $Q_{pe}/P$), we find that the event water fraction of precipitation ($Q_e/P$) correlates strongly with metrics of storm characteristics (but not antecedent wetness), and the ratio of pre-event water to precipitation ($Q_{pe}/P$) correlates strongly with metrics of antecedent wetness (but not storm characteristics). Thus, $Q_e/P$ and $Q_{pe}/P$ more clearly reflect the influence of major controls on streamflow
generation compared to $Q_e/Q$ (or $Q_{pe}/Q$).

3. Although numerous studies have used runoff coefficients or tracer-based event water fractions $Q_e/Q$ to study catchment hydrological behavior during storm events (e.g., Klaus and McDonnell, 2013), the additional information provided by $Q_e/P$ and $Q_{pe}/P$ has yet not been exploited. Together, $Q_e/P$ and $Q_{pe}/P$ separate the runoff coefficient $Q/P$ into its contributions from event water and pre-event water, providing a straightforward way to quantify both of these
components of streamflow response.

4. At Erlenbach, $Q/P$, $Q_e/P$ and $Q_{pe}/P$ exhibit roughly linear relationships with several measures of storm size and antecedent wetness, suggesting that these relationships, and particularly their slopes, may be diagnostic "fingerprints" that may be useful for characterizing hydrologic response across diverse catchments.

5. Looking toward the future, we anticipate that hydrograph separation studies will increasingly seek to quantify many
different sources of streamflow, beyond the traditional separation of $Q_e$ and $Q_{pe}$. We note that if other components can be identified (e.g., streamflow originating from throughfall, soil water, deep bedrock water, etc.), studying their volumes relative to precipitation, rather than discharge, may shed important light on how they are mobilized during storm events.

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





**Tables**

**Table 1: Properties of the storm events: total discharge ($Q$), total precipitation ($P$), cumulative precipitation before peak flow ($P_{untilQpeak}$), mean precipitation intensity ($P_{int}$), maximum precipitation over 1 hour ($P_{1h}$) and 4 hours ($P_{4h}$), event duration ($T$), and four-hour peak discharge volume ($Q_{peak}$). The initial catchment wetness state was quantified using three-day and seven-day antecedent precipitation ($AP3$ and $AP7$), as well as the 1-hour average values of discharge ($Q_{ini}$), groundwater table depth ($GW_{ini}$), and soil moisture ($SM_{ini}$) before the onset of the storm event.**

| Event | $Q$ (mm) | $P$ (mm) | $P_{untilQpeak}$ (mm) | $P_{1h}\pm SE$ (mm) | $P_{4h}\pm SE$ (mm) | $T\pm 1h$ (h) | $P_{int}$ (mm h$^{-1}$) | $Q_{peak}\pm SE$ (mm) | $AP3$ (mm) | $AP7$ (mm) | $Q_{ini}\pm SE$ (mm h$^{-1}$) | $SM_{ini}$ (m$^3$ m$^{-3}$) | $GW_{ini}\pm SE$ (cm below arbitrary height) |
|---|---|---|---|---|---|---|---|---|---|---|---|---|---|
| 17Sep2016 | 0.8 | 9.2 | 8.6 | 6.1±1.1 | 8.4±4.0 | 7.5 | 1.23 | 0.25±0.00 | 7.3 | 7.3 | 0.018±0.000 | 0.419 | 47.3±0.1 |
| 02Oct2016 | 4.8 | 21.6 | 21.4 | 5.4±0.1 | 11.5±3.0 | 15.5 | 1.39 | 1.23±0.10 | 0.7 | 10.8 | 0.012±0.000 | 0.427 | 33.4±0.2 |
| 09Oct2016 | 0.4 | 12.6 | 12.5 | 9.1±1.7 | 12.5±5.6 | 2.5 | 5.04 | 0.11±0.00 | 0.4 | 9.4 | 0.012±0.000 | 0.428 | 30.2±0.2 |
| 13May2017 | 9.5 | 20.8 | 20.2 | 9.0±1.6 | 20.4±6.2 | 5.5 | 3.78 | 6.11±0.62 | 16.7 | 91.2 | 0.200±0.000 | 0.444 | 17.3±0.1 |
| 19May2017 | 5.3 | 17.6 | 17.6 | 4.2±0.1 | 11.0±2.8 | 7.0 | 2.51 | 1.48±0.08 | 1.2 | 43.6 | 0.063±0.000 | 0.433 | 28.4±0.1 |
| 01Jun2017 | 7.5 | 28.5 | 26.9 | 23.2±6.5 | 27.7±17.4 | 6.5 | 4.38 | 4.98±0.65 | 20.6 | 20.6 | 0.042±0.000 | 0.433 | 43.0±0.2 |
| 04Jun2017 | 17.4 | 33.7 | 30.3 | 10.8±0.6 | 26.2±6.5 | 11.5 | 2.93 | 10.57±1.62 | 43.1 | 60.3 | 0.074±0.001 | 0.439 | 20.1±0.1 |
| 07Jun2017 | 8.0 | 11.2 | 11.2 | 3.7±0.8 | 6.5±2.5 | 6.5 | 1.72 | 2.53±0.16 | 50.3 | 109.4 | 0.278±0.005 | 0.445 | 16.3±0.2 |
| 16Jun2017 | 17.1 | 46.0 | 39.9 | 18.1±7.7 | 25.8±12.4 | 9.5 | 4.83 | 9.70±1.31 | 0 | 2.5 | 0.014±0.000 | 0.424 | 39.4±0.1 |
| 25Jun2017 | 1.7 | 21.2 | 21.2 | 15.2±0.9 | 16.3±9.9 | 6.5 | 3.26 | 1.01±0.14 | 1.4 | 1.4 | 0.013±0.000 | 0.414 | 48.4±0.1 |
| 10Jul2017 | 7.0 | 25.4 | 25.4 | 21.4±6.8 | 23.0±17.1 | 6.8 | 3.76 | 5.30±0.93 | 20.2 | 20.2 | 0.032±0.000 | 0.437 | 35.3±0.2 |
| 19Jul2017 | 4.0 | 20.2 | 18.1 | 10.3±1.1 | 18.1±6.7 | 10.5 | 1.92 | 1.58±0.18 | 0.5 | 16.8 | 0.016±0.000 | 0.427 | 31.0±0.1 |
| 27Jul2017 | 7.7 | 12.9 | 12.7 | 6.8±0.5 | 7.1±4.5 | 11.0 | 1.17 | 1.49±0.06 | 74.4 | 105.8 | 0.109±0.001 | 0.438 | 21.6±0.2 |
| 05Aug2017 | 2.7 | 17.1 | 15.8 | 8.5±0.1 | 15.8±10.4 | 9.5 | 1.80 | 1.30±0.18 | 0 | 15.8 | 0.020±0.000 | 0.429 | 29.7±0.1 |
| 06Aug2017 | 4.8 | 11.2 | 10.8 | 5.7±0.3 | 9.7±3.2 | 9.5 | 1.18 | 1.86±0.12 | 17.4 | 32.6 | 0.084±0.001 | 0.443 | 17.6±0.2 |
| 15Aug2017 | 2.3 | 8.2 | 8.2 | 3.7±1.6 | 4.5±0.3 | 7.5 | 1.09 | 0.37±0.00 | 0.6 | 86.5 | 0.042±0.000 | 0.433 | 25.5±0.2 |
| 18Aug2017 | 25.6 | 56.3 | 34.7 | 31.0±5.3 | 43.6±19.6 | 12.0 | 4.68 | 11.71±1.64 | 8.5 | 26.3 | 0.028±0.000 | 0.431 | 28.5±0.2 |
| 12Sep2017 | 9.2 | 19.7 | 18.9 | 5.8±0.1 | 10.8±3.3 | 11.5 | 1.71 | 2.76±0.16 | 45.5 | 53 | 0.082±0.000 | 0.440 | 21.3±0.2 |
| 25Sep2017 | 3.5 | 15.2 | 15.2 | 14.9±5.5 | 15.3±12.7 | 1.5 | 10.12 | 1.93±0.25 | 0.2 | 20.4 | 0.026±0.000 | 0.434 | 27.6±0.2 |
| 02Oct2017 | 20.9 | 39.1 | 37.9 | 10.9±0.3 | 23.6±5.5 | 19.5 | 2.00 | 12.50±1.68 | 28 | 48 | 0.059±0.000 | 0.440 | 19.8±0.2 |
| 05Oct2017 | 20.5 | 33.5 | 31.9 | 5.3±0.8 | 19.5±1.9 | 9.5 | 3.52 | 7.99±0.65 | 39.2 | 67.9 | 0.047±0.000 | 0.438 | 22.0±0.2 |
| 22Oct2017 | 25.5 | 63.2 | 59.4 | 8.1±0.3 | 18.7±4.1 | 51.6 | 1.23 | 2.14±0.05 | 0 | 0 | 0.013±0.000 | 0.429 | 36.5±0.1 |
| 26Oct2017 | 4.3 | 12.5 | 12.3 | 6.1±0.6 | 10.5±3.5 | 6.5 | 1.92 | 1.85±0.15 | 0 | 63.2 | 0.038±0.001 | 0.437 | 22.5±0.2 |
| 29Oct2018 | 25.7 | 44.8 | 27.2 | 8.4±0.3 | 22.8±3.0 | 17.0 | 2.64 | 10.19±0.74 | 12.5 | 56 | 0.053±0.000 | 0.441 | 19.3±0.1 |





**Table 2: Results of the hydrograph separation based on $\delta^2$H (results for $\delta^{18}$O are provided in Table S1 in the Supplement). Columns are total discharge ($Q$), total precipitation ($P$), the runoff coefficient ($Q/P$), event and pre-event discharge as whole-storm totals ($Q_e$ and $Q_{pe}$), the maximum instantaneous event water fraction $q_{e,i}/q_i$ and its value at peak flow, the event water fractions of discharge and precipitation ($Q_e/Q$ and $Q_e/P$), and pre-event discharge as a fraction of precipitation ($Q_{pe}/P$).**

| Event | $Q$ (mm) | $P$ (mm) | $Q/P$ (-) | $Q_e\pm SE$ (mm) | $Q_{pe}\pm SE$ (mm) | $\max(q_{e,i}/q_i)\pm SE$ (-) | $q_{e,i}/q_i$ at peak flow$\pm SE$ (-) | $Q_e/Q\pm SE$ (-) | $Q_e/P\pm SE$ (-) | $Q_{pe}/P\pm SE$ (-) |
|---|---|---|---|---|---|---|---|---|---|---|
| 17Sep2016 | 0.8 | 9.2 | 0.08 | 0.30±0.02 | 0.45±0.02 | 0.51±0.05 | 0.45±0.04 | 0.40±0.03 | 0.033±0.002 | 0.049±0.002 |
| 02Oct2016 | 4.8 | 21.6 | 0.22 | 1.09±0.04 | 3.68±0.04 | 0.30±0.01 | 0.30±0.01 | 0.23±0.01 | 0.051±0.002 | 0.170±0.002 |
| 09Oct2016 | 0.4 | 12.6 | 0.03 | 0.03±0.00 | 0.41±0.00 | 0.10±0.01 | 0.06±0.01 | 0.07±0.00 | 0.002±0.000 | 0.033±0.000 |
| 13May2017 | 9.5 | 20.8 | 0.46 | 1.50±0.10 | 7.97±0.10 | 0.31±0.06 | 0.21±0.04 | 0.16±0.01 | 0.072±0.005 | 0.383±0.005 |
| 19May2017 | 5.3 | 17.6 | 0.30 | 0.27±0.02 | 5.03±0.02 | 0.11±0.03 | 0.08±0.01 | 0.05±0.00 | 0.015±0.001 | 0.286±0.001 |
| 01Jun2017 | 7.5 | 28.5 | 0.26 | 1.88±0.07 | 5.66±0.07 | 0.48±0.02 | 0.30±0.02 | 0.25±0.01 | 0.066±0.002 | 0.199±0.002 |
| 04Jun2017 | 17.4 | 33.7 | 0.52 | 2.26±0.11 | 15.13±0.11 | 0.17±0.02 | 0.14±0.02 | 0.13±0.01 | 0.067±0.003 | 0.449±0.003 |
| 07Jun2017 | 8.0 | 11.2 | 0.72 | 0.41±0.01 | 7.63±0.01 | 0.10±0.01 | 0.08±0.01 | 0.05±0.00 | 0.037±0.001 | 0.682±0.001 |
| 16Jun2017 | 17.1 | 46.0 | 0.37 | 7.13±0.57 | 9.94±0.57 | 0.74±0.13 | 0.46±0.06 | 0.42±0.03 | 0.155±0.012 | 0.216±0.012 |
| 25Jun2017 | 1.7 | 21.2 | 0.08 | 0.35±0.00 | 1.37±0.00 | 0.26±0.01 | 0.26±0.01 | 0.20±0.00 | 0.016±0.000 | 0.065±0.000 |
| 10Jul2017 | 7.0 | 25.4 | 0.28 | 3.42±0.11 | 3.57±0.11 | 0.57±0.02 | 0.56±0.03 | 0.49±0.02 | 0.135±0.005 | 0.141±0.005 |
| 19Jul2017 | 4.0 | 20.2 | 0.20 | 1.12±0.02 | 2.91±0.02 | 0.35±0.01 | 0.35±0.01 | 0.28±0.00 | 0.055±0.001 | 0.144±0.001 |
| 27Jul2017 | 7.7 | 12.9 | 0.59 | 0.57±0.01 | 7.10±0.01 | 0.13±0.01 | 0.10±0.01 | 0.07±0.00 | 0.044±0.001 | 0.550±0.001 |
| 05Aug2017 | 2.7 | 17.1 | 0.16 | 0.52±0.02 | 2.15±0.02 | 0.26±0.03 | 0.23±0.03 | 0.20±0.01 | 0.031±0.001 | 0.125±0.001 |
| 06Aug2017 | 4.8 | 11.2 | 0.43 | 0.46±0.01 | 4.31±0.01 | 0.16±0.01 | 0.15±0.01 | 0.10±0.00 | 0.041±0.001 | 0.385±0.001 |
| 15Aug2017 | 2.3 | 8.2 | 0.28 | 0.10±0.00 | 2.22±0.00 | 0.08±0.01 | 0.06±0.01 | 0.04±0.00 | 0.012±0.000 | 0.270±0.000 |
| 18Aug2017 | 25.6 | 56.3 | 0.45 | 19.25±1.03 | 6.33±1.03 | 0.84±0.02 | 0.84±0.04 | 0.75±0.04 | 0.342±0.018 | 0.112±0.018 |
| 12Sep2017 | 9.2 | 19.7 | 0.46 | 0.33±0.03 | 8.82±0.03 | 0.13±0.02 | 0.03±0.01 | 0.04±0.00 | 0.017±0.001 | 0.448±0.001 |
| 25Sep2017 | 3.5 | 15.2 | 0.23 | 0.73±0.01 | 2.78±0.01 | 0.36±0.01 | 0.26±0.01 | 0.21±0.00 | 0.048±0.001 | 0.183±0.001 |
| 02Oct2017 | 20.9 | 39.1 | 0.53 | 5.62±0.69 | 15.25±0.69 | 0.31±0.08 | 0.31±0.05 | 0.27±0.03 | 0.144±0.018 | 0.390±0.018 |
| 05Oct2017 | 20.5 | 33.5 | 0.61 | 5.03±0.11 | 15.52±0.11 | 0.33±0.01 | 0.30±0.01 | 0.24±0.01 | 0.150±0.003 | 0.463±0.003 |
| 22Oct2017 | 25.5 | 63.2 | 0.40 | 11.32±0.32 | 14.17±0.32 | 0.58±0.01 | 0.56±0.02 | 0.44±0.01 | 0.179±0.005 | 0.224±0.005 |
| 26Oct2017 | 4.3 | 12.5 | 0.34 | 0.85±0.03 | 3.40±0.03 | 0.27±0.02 | 0.26±0.02 | 0.20±0.01 | 0.068±0.002 | 0.272±0.002 |
| 29Oct2017 | 25.7 | 44.8 | 0.57 | 8.88±0.17 | 16.85±0.17 | 0.42±0.01 | 0.41±0.01 | 0.35±0.01 | 0.198±0.004 | 0.376±0.004 |





**Table 3: Spearman rank correlation coefficients ($\rho$) and $p$-values for measures of storm characteristics, antecedent wetness, and catchment storm response. Fields with dark grey background represent statistically significant correlations with $p<0.0001$, and fields with light grey background represent statistically significant correlations with $p<0.01$. Measures of storm characteristics (left-hand columns in table) are total event precipitation ($P$), cumulative precipitation before peak flow ($P_{\text{untilQpeak}}$), mean precipitation intensity ($P_{\text{int}}$), maximum precipitation over 1 hour ($P_{1h}$) and 4 hours ($P_{4h}$), event duration ($T$), and four-hour peak discharge volume ($Q_{\text{peak}}$). Measures of initial catchment wetness state (right-hand columns in table) are by three-day and seven-day antecedent precipitation ($AP3$ and $AP7$), as well as the 1-hour average values of discharge ($Q_{\text{ini}}$), groundwater table depth ($GW_{\text{ini}}$), and soil moisture ($SM_{\text{ini}}$) before the onset of the storm event. Measures of catchment storm response (rows of table) are precipitation ($P$), discharge ($Q$), event discharge ($Q_e$), and pre-event discharge ($Q_{\text{pe}}$), all defined as totals over the event, and ratios among them.**

| $\rho$ | $P$ | $P_{\text{untilQpeak}}$ | $P_{1h}$ | $P_{4h}$ | $P_{\text{int}}$ | $T$ | $AP3$ | $AP7$ | $Q_{\text{ini}}$ | $SM_{\text{ini}}$ | $GW_{\text{ini}}$ |
|---|---|---|---|---|---|---|---|---|---|---|---|
| $P$ | *1.00* | *0.99* | 0.55 | *0.88* | 0.42 | **0.57** | 0.03 | -0.25 | -0.21 | -0.06 | -0.17 |
| $Q$ | *0.75* | *0.74* | 0.14 | **0.57** | 0.13 | **0.60** | 0.41 | 0.33 | 0.39 | 0.50 | 0.41 |
| $Q_e$ | *0.87* | *0.85* | 0.50 | *0.79* | 0.32 | **0.53** | 0.04 | -0.05 | -0.07 | 0.13 | 0.01 |
| $Q_{\text{pe}}$ | *0.65* | *0.67* | -0.03 | 0.45 | 0.04 | **0.57** | 0.48 | 0.41 | 0.49 | **0.58** | 0.50 |
| $Q/P$ | 0.27 | 0.29 | -0.26 | 0.09 | -0.14 | 0.39 | *0.65* | *0.74* | *0.76* | *0.80* | *0.76* |
| $Q_e/Q$ | *0.65* | **0.62** | 0.59 | **0.64** | 0.31 | 0.34 | -0.24 | -0.51 | -0.50 | -0.35 | -0.49 |
| $Q_e/P$ | *0.78* | *0.76* | 0.41 | *0.71* | 0.31 | 0.48 | 0.04 | 0.03 | -0.01 | 0.20 | 0.08 |
| $Q_{\text{pe}}/P$ | 0.00 | 0.05 | -0.46 | -0.15 | -0.30 | 0.22 | *0.63* | *0.79* | *0.83* | *0.81* | *0.79* |

| $p$ | $P$ | $P_{\text{untilQpeak}}$ | $P_{1h}$ | $P_{4h}$ | $P_{\text{int}}$ | $T$ | $AP3$ | $AP7$ | $Q_{\text{ini}}$ | $SM_{\text{ini}}$ | $GW_{\text{ini}}$ |
|---|---|---|---|---|---|---|---|---|---|---|---|
| $P$ | *0.0000* | *0.0000* | 0.0055 | *0.0000* | 0.0407 | **0.0037** | 0.8955 | 0.2336 | 0.3349 | 0.7713 | 0.4186 |
| $Q$ | *0.0000* | *0.0000* | 0.5126 | **0.0039** | 0.5490 | **0.0019** | 0.0487 | 0.1210 | 0.0580 | 0.0120 | 0.0479 |
| $Q_e$ | *0.0000* | *0.0000* | 0.0129 | *0.0000* | 0.1220 | **0.0073** | 0.8604 | 0.8150 | 0.7436 | 0.5408 | 0.9678 |
| $Q_{\text{pe}}$ | *0.0006* | *0.0004* | 0.9053 | 0.0292 | 0.8401 | **0.0038** | 0.0182 | 0.0439 | 0.0160 | **0.0029** | 0.0130 |
| $Q/P$ | 0.2064 | 0.1753 | 0.2269 | 0.6654 | 0.5036 | 0.0626 | *0.0006* | *0.0000* | *0.0000* | *0.0000* | *0.0000* |
| $Q_e/Q$ | *0.0006* | **0.0014** | **0.0024** | *0.0007* | 0.1352 | 0.1029 | 0.2647 | 0.0115 | 0.0128 | 0.0941 | 0.0156 |
| $Q_e/P$ | *0.0000* | *0.0000* | 0.0440 | *0.0001* | 0.1470 | 0.0181 | 0.8477 | 0.8877 | 0.9582 | 0.3552 | 0.7193 |
| $Q_{\text{pe}}/P$ | 0.9871 | 0.8181 | 0.0221 | 0.4702 | 0.1482 | 0.2917 | *0.0009* | *0.0000* | *0.0000* | *0.0000* | *0.0000* |





**Figures**

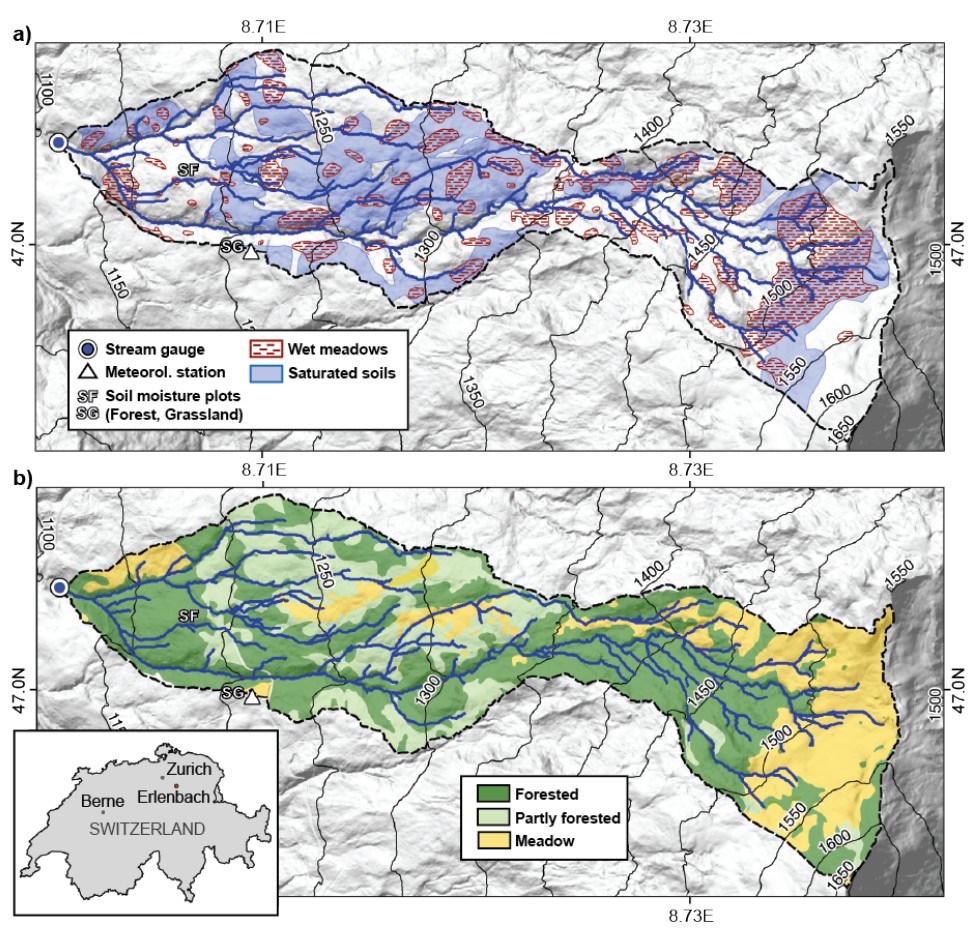

**Figure 1: Th Erlenbach catchment with spatial distribution of (a) saturated soils (Swiss Federal Office for the Environment (FOEN), 2011) and mapped wet meadows; (b) vegetation (Fischer et al., 2015).**





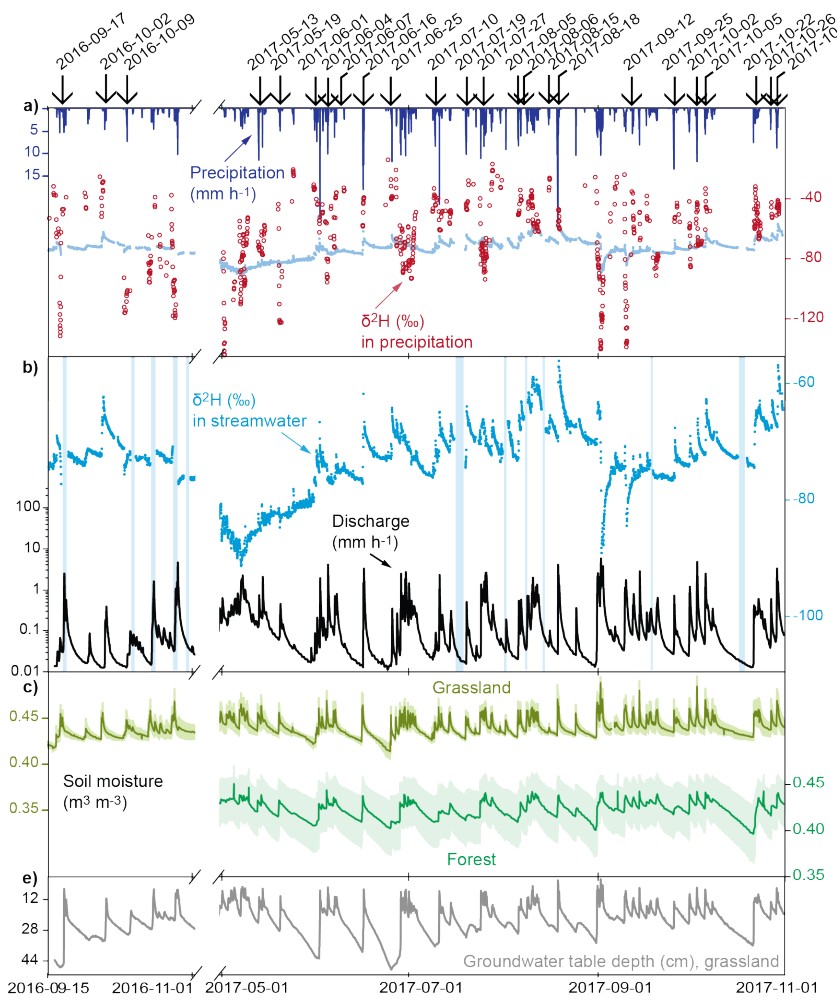

**Figure 2: Time series of a) precipitation and δ²H in precipitation (δ²H in streamwater is shown for comparison); b) discharge (on log axis) and δ²H in streamwater (vertical bars indicate gaps in the isotope data); c) soil moisture measured at 50 cm depth at the grassland and forest sites (Fig. 1a), with shaded areas showing the standard error from averaging the measurements from the four probes at each plot; and d) groundwater levels at the grassland site. The winter period with snow cover (06Nov2016-05May2017) was not considered in this analysis since the individual contributions of rainfall and snowmelt to river discharge could not be distinguished sufficiently. Vertical arrows indicate the events analysed in this paper.**





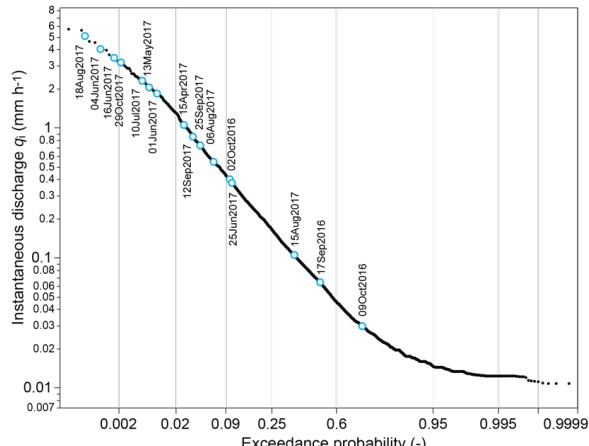

**Figure 3: Flow duration curve of Erlenbach for the study periods 15Sep2016 to 06Nov2016 and 05May2017 to 01Nov2017. Blue data points indicate the peak flow rates of some storm events analysed in this study, showing that widely varying flow conditions were captured in our data set.**



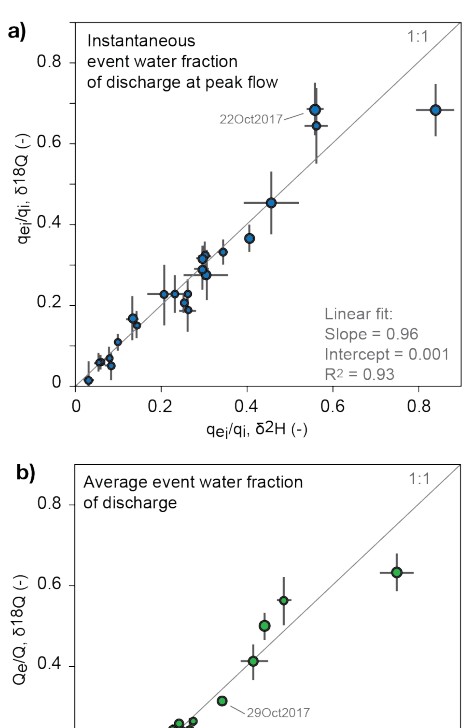

**Figure 4: a)** Instantaneous event water fractions of discharge at peak flow ($q_{e,i}/q_i$), as well as **b)** whole-storm event water fractions of discharge ($Q_e/Q$) obtained from either $\delta^2H$ or $\delta^{18}O$. The sizes of the data points reflect precipitation totals of the storm events and error bars show $\pm 1$ *SE*. For the 17Sep2017 storm, unrealistic results were obtained for $Q_e$ when $\delta^{18}O$ was used as a tracer, and therefore this data point is excluded from the comparison analysis. Hydrograph separation results for $\delta^{18}O$ are provided Table S1 in the Supplement.





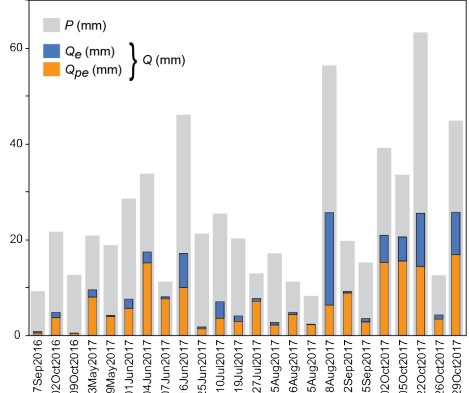

**Figure 5: Volumes of precipitation (*P*, grey), compared to event water (*Q*ₑ, blue) and pre-event water (*Q*ₚₑ ,orange) in discharge across 24 storm events. Total discharge (*Q*) is the sum of event and pre-event water. For most of the storms, pre-event water comprised the major fraction of streamflow. Event water dominated streamflow only during two storms (10Jul2017 and 18Aug2017).**



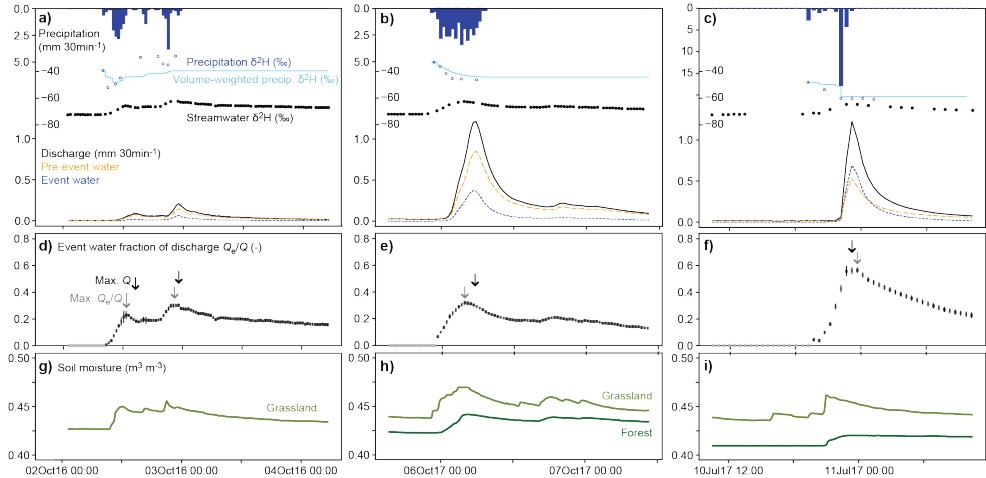

**Figure 6: Time series of the storm events of 02Oct2016, 05Oct2017 and 10Jul2017.** (a - c) Precipitation hyetographs and deuterium abundance ($\delta^2$H) in precipitation (with individual measurements in dark blue and incremental weighted means in light blue), and deuterium streamwater (black), as well as discharge hydrograph separated into event and pre-event water. (d - f) Event water fraction of discharge; error bars indicate ±1*SE*, and open bars indicate linearly interpolated event water fractions when discharge isotope measurements are missing. (g - i) Soil moisture at the grassland site (light green) and forest site (dark green, no data in 2016). Despite great differences in total event rainfall and antecedent wetness conditions between the two storms, in the two left columns, their event water fractions of discharge are very similar. In most cases, peaks in instantaneous event water fractions precede peak flows (times of peak values indicated by grey and black vertical arrows in panels a to f)



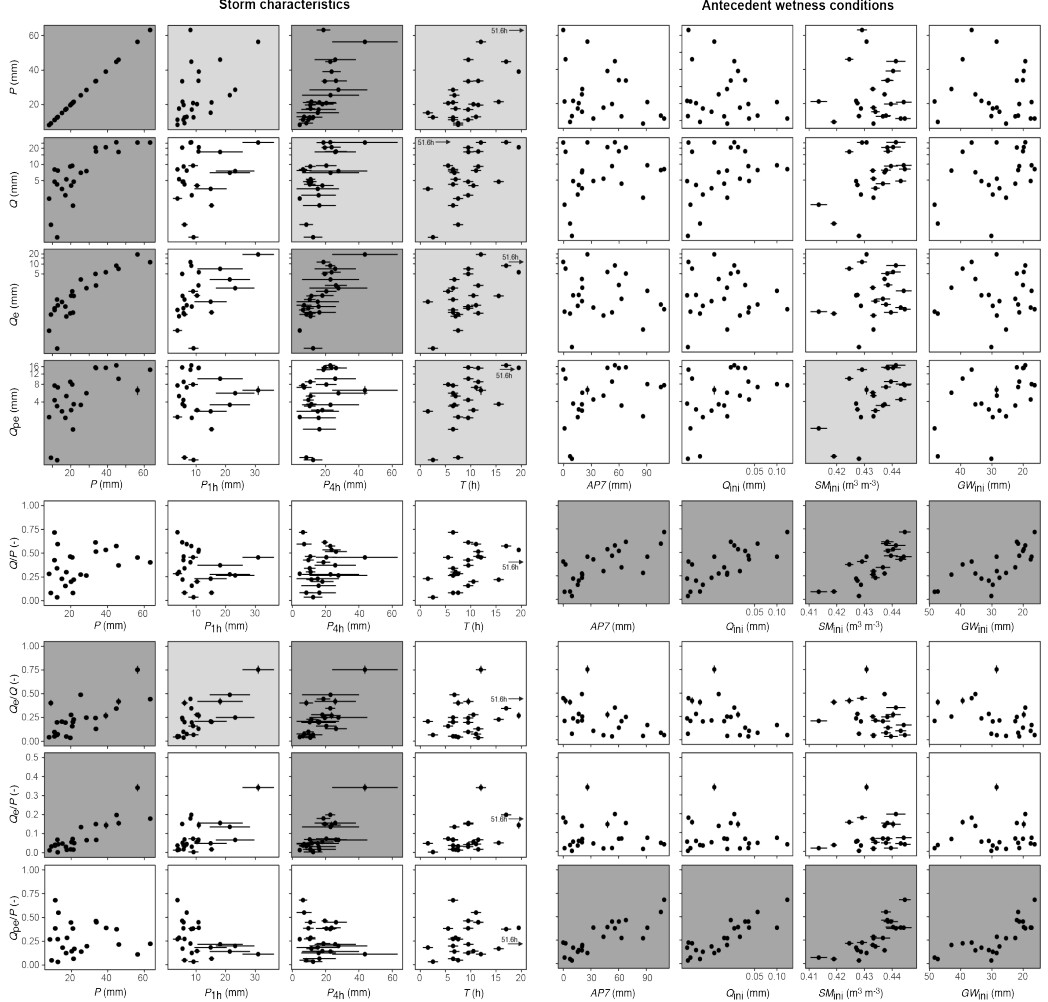

**Figure 7: Total volumes of storm precipitation, discharge, event and pre-event water ($Q_e$, $Q_{pe}$), backward event- and pre-event water fractions ($Q_e/Q$, $Q_{pe}/Q$), as well as forward event- and pre-event water fractions ($Q_e/P$, $Q_{pe}/P$) of the 24 storm events, plotted against metrics of storm characteristics and catchment antecedent wetness conditions. Measures of storm characteristics (left-hand columns) are total, peak 1-hour, and peak 4-hour precipitation ($P$, $P_{1h}$, $P_{4h}$), and total event duration ($T$). Measures of antecedent wetness (right-hand columns) are seven-day antecedent precipitation ($AP7$) and the 1-hour average values of discharge ($Q_{ini}$), soil moisture ($SM_{ini}$), and groundwater table depth ($GW_{ini}$) before the onset of the storm event. Panels with light grey backgrounds indicate correlations that are statistically significant at $p<0.01$; panels with dark grey backgrounds indicate**





correlations that are statistically significant at $p<0.0001$. Because the precipitation duration ($T$) of storm 22Oct2017 was 51.6 hours, it is off-scale and is thus indicated with vertical arrows.