# Peer review of "Studying catchment storm response using event and pre-event water volumes as fractions of precipitation rather than discharge"

_Hydrology and Earth System Sciences, 2018_

## Referee Comment (RC1) · D. Penna (Referee) · 22 Sep 2018

Review of the manuscript 'Studying catchment storm response using event and pre-event water volumes as fractions of precipitation rather than discharge' by von Freyberg et al. (hess-2018-401)

General comment I read the manuscript by von Freyberg and colleagues with keen interest. They use high-resolution stable isotope data of stream water and precipitation collected during 24 rainfall-runoff events in a small Swiss catchment to test the usefulness of an alternative metric for studying runoff generation processes at the catchment scale. They argue that the commonly adopted tracer-based estimates of event and

pre-event water fractions of stream runoff (Qe/Q, Qpe/Q), typically used to analyze the fundamental controls on catchment hydrological response, may be ambiguous because the same controls on Qe and Qpe also necessary control the total discharge Q. Therefore, the authors suggest using the fraction of event and pre-event water relative to precipitation, instead (Qe/P and Qpe/P), asserting that it may provide an alternative and more insightful approach analyze catchment hydrological response.

The authors support their thoughts with clear field-based evidence and produce convincing results showing the effectiveness of this alternative metric to reveal runoff generation processes, at least in the study catchment. This study contains a high degree of novelty, and constitutes a scientific advancement in catchment hydrological science as it can open up new ways to take the best advantage of the more and more widely adopted stable isotopes in water to investigate hydrological processes at the catchment scale.

Overall, the manuscript is very well written, logically organized, and clearly illustrated. All methods are clearly described, the authors' thoughts can be followed effortlessly, and the results are solidly supported by the data. There are some parts where a certain degree of redundancy exists but this does not hurt and may even help stress some relevant points. I have only a few comments to improve the manuscript, and I recommend a minor revision before publication.

Specific comments P1L9-10, and P1L18. Here, and in other parts of the manuscript, I suggest specifying that this work focuses on the two-component hydrograph separation, which is used to estimate the pre-event and event component of stream runoff. Indeed, the word "source", used, for instance, at P1L18, P2L17 is, in my opinion, a bit vague: one tracer (often isotopes), two-component hydrograph separation is typically used to estimate *time* source components of total discharge, whereas two-tracer (or more than two tracers, usually isotopes plus hydrochemical tracers), three (or multi-) component hydrograph separation is often used to estimate *geographical* source components (Klaus and McDonnell, 2013), such as snowmelt, glacier melt, or hillslope

soil water, riparian soil water, shallow groundwater etc. I think that this specification and distinction should be made clear in the abstract and throughout the paper (not only in the title of Section 2.4).

P3L10. This is true and was shown in several studies. However, also the opposite can happen, depending on the specific catchment properties. For instance, in our Dolomitic experimental catchment tracer data showed that, on average, Qe increases with increasing antecedent moisture conditions mainly due to the streamflow contribution of saturation overland flow, formed by a mixture of rain water falling on the saturated areas and pre-event water exfiltrating in the riparian zone (Penna et al., 2016). This may occur also in other catchments.

Penna, D., van Meerveld, H.J., Zuecco, G., Dalla Fontana, G., Borga, M., 2016. Hydrological response of an Alpine catchment to rainfall and snowmelt events. Journal of Hydrology 537, 382–397. https://doi.org/10.1016/j.jhydrol.2016.03.040

P4L15-20. The general aim of the study is clear but I suggest formulating here specific objectives and/or a clear testable hypothesis.

P7, Section 2.3. I think that the selection of metrics used to characterize the storm properties and the antecedent wetness conditions are appropriate. However, I think it would be interesting to add the combination of P and SMini as a metric (e.g.,Detty and McGuire, 2010; Fu et al., 2013) and see if and how the fractions of Qe/Q, Qpe/Q, and most of all Qe/P and Qpe/P are sensitive to it.

Detty JM, McGuire KJ. 2010. Threshold changes in storm runoff generation at a till-mantled headwater catchment. Water Resources Research 46: W07525. DOI: 10.1029/2009WR008102

Fu C, Cheng J, Jiang H, Dong L. 2013. Threshold behavior in a fissured granitic catchment in southern China: (1) analysis of field monitoring results. Water Resources Research 49: 1–17. DOI: 10.1002/wrcr.20191

P18L11. I agree but AP7 (and AP indices in general) is only a surrogate of the catchment antecedent wetness status (Ali and Roy, 2010) and therefore this relation could not be robust and reliable "fingerprint". Maybe a sentence on this could be added.

Ali, G. A. and Roy, A. G.: A case study on the use of appropriate surrogates for antecedent moisture conditions (AMCs), Hydrol. Earth Syst. Sci., 14, 1843-1861, https://doi.org/10.5194/hess-14-1843-2010, 2010.

Minor comments and technical corrections P1L9. It is not immediately clear if the terms "streamflow" and "discharge" are used interchangeably or if they imply a different meaning. In the first case, I suggest to use one term consistently. In the second case, I suggest to indicate the possible distinction.

P5L3. I suggest replacing "soil depths are shallower" with "soils are shallower".

P8L6. So, in the last 2.5 hours?

P10L21. Please, report the p-value here as well.

P12L7. "river": earlier in the manuscript the authors used the term "streamwater" (eg, P7L17), so I imagine (also considering the catchment size) that the term "stream" is more appropriate here.

P14L2-3. As far as I understand, the authors here mean "but are not, however, statistically significant (p>0.01)" or "but are, however, statistically not significant (p>0.01)".

P14L26. I suggest adding a reference here. Examples might be McGlynn and McDonnell (2003), James and Roulet (2009), Muñoz-Villers and McDonnell (2012).

James AL, Roulet NT. 2009. Antecedent moisture conditions and catchment morphology as controls on spatial patterns of runoff generation in small forest catchments. Journal of Hydrology 377(3-4): 351–366. DOI: 10.1016/j.jhydrol.2009.08.039

McGlynn, B. L., and J. J. McDonnell (2003), Quantifying the relative contributions of riparian and hillslope zones to catchment runoff, Water Resour. Res., 39, 1310, doi:

[Figure]

10.1029/2003WR002091, 11.

Muñoz-Villers LE, McDonnell JJ. 2012. Runoff generation in a steep, tropical montane cloud forest catchment on permeable volcanic substrate, Water Resources Research 48: W09528. DOI: 10.1029/ 2011WR011316

P14L28. "which sum to the runoff coefficient itself: Q/P=Qe/P+Qpe/P". This has been said more than once before, and can be dropped.

P15L3. I suggest replacing "tightly" with "strongly".

P18L18. I suggest replacing "forever" with "also for large values" or something similar.

Table 2. I suggest dropping the second and the third column (Q and P) because already reported in Table 1. This can improve the readability of the table.

Fig. 2. The label of the last panel should be "d)" and not "e)".

Fig. 3. So, if I understand well, this flow duration curve a combination of two distinct periods. I wonder whether it would be more appropriate to show two curves for the two periods separately.

Fig. 4. In the label of the two y-axis correct "18Q" with "18O".

Fig. 7. The caption is not complete.

---

## Referee Comment (RC2) · Anonymous Referee #2 · 10 Oct 2018

Review for manuscript Manuscript ID: HESS-2018-401; Studying catchment storm response using event and pre-event water volumes as fractions of precipitation rather than discharge

Best authors and editors, Thank you for the possibility to review this very interesting manuscript, and apologies for the delay in my review.

The authors present a hydrograph separation studying the stream water sources in an experimental Erlenbach catchment in Switzerland. The work builds on an advanced field laboratory, enabling high-frequency determination of isotope composition in stream water and precipitation used in identifying pre-event and event water com-

position, respectively. Authors present an eight-month long dataset of isotope and hydrometric measurements for flow and precipitation, supplemented with groundwater level and soil moisture data as proxies for catchment wetness. As a subset of this data, they analyse 24 storms in greater detail. The results show the advantages in exploring the pre-event and event water contributions as a fraction of precipitation, not total streamflow as is typically done. Using this approach, the authors were able to infer novel insights to catchment controls on streamflow generation. I particularly enjoyed section "3.4 fingerprints of catchment response" in which the authors put forward interesting hypothesis to be tested by the hydrological community.

The manuscript is written with flawless English, and is well structured and presented. In my opinion both the collected and dataset and the following analysis are novel and of high quality, and therefore a great contribution to the hydrological sciences. I recommend this work to be published in HESS, and provide some minor remarks below.

comments: P4L13: I would recommend the authors to better acknowledge and discuss prior work studying the Qe/P ratio in the introduction. Before this chapter, I had the impression this is being done the first time in the presented manuscript. P5L5 what do you mean by "saturated soils"? groundwater table is at ground level? Or that the soil type is prone to saturation? I presume that the extent of saturation would vary seasonally, so a static map for it seems simplified. Fig.1: add a scale, the degree axis are not very intuitive of the catchment size P7L16: concentrations -> ratios? P8L10: how is Q for each event defined and calculated? P10L1: add spacing for dates in all occurences? P10 L15: I don't understand how the 4-hour peak Q 0.11 mm is lower that overall Q 0.5 mm. How is 4-hour peak Q defined? Fig. 4: should y-axis be delta 18O? P10L 24: how about the point on the far right in both a) and b) plots? That deviates substantially from the 1:1 line. P15L10 and table3: I don't find Qpe/Q data in table 3, though discussed in the text P17L6: I see this conclusion somewhat inconsistent with your data analysis so far. You suggest that the Pe could be explained by contraction and expansion of saturated areas, i.e. the antecedent conditions, whereas before you

demonstrate and discuss how the Pe is mainly a function of the storm characteristics.

---

## Author Comment (AC1) · 11 Oct 2018

**Response to the interactive comment of D. Penna**

on "Studying catchment storm response using event and pre-event water volumes as fractions of precipitation rather than discharge" by Jana von Freyberg et al.

**General comment**

*I read the manuscript by von Freyberg and colleagues with keen interest. They use high-resolution stable isotope data of stream water and precipitation collected during 24 rainfall-runoff events in a small Swiss catchment to test the usefulness of an alternative metric for studying runoff generation processes at the catchment scale. They argue that the commonly adopted tracer-based estimates of event and pre-event water fractions of stream runoff (Qe/Q, Qpe/Q), typically used to analyze the fundamental controls on catchment hydrological response, may be ambiguous because the same controls on Qe and Qpe also necessary control the total discharge Q. Therefore, the authors suggest using the fraction of event and pre-event water relative to precipitation, instead (Qe/P and Qpe/P), asserting that it may provide an alternative and more insightful approach analyze catchment hydrological response. The authors support their thoughts with clear field-based evidence and produce convincing results showing the effectiveness of this alternative metric to reveal runoff generation processes, at least in the study catchment.*

*This study contains a high degree of novelty, and constitutes a scientific advancement in catchment hydrological science as it can open up new ways to take the best advantage of the more and more widely adopted stable isotopes in water to investigate hydrological processes at the catchment scale.*

*Overall, the manuscript is very well written, logically organized, and clearly illustrated. All methods are clearly described, the authors' thoughts can be followed effortlessly, and the results are solidly supported by the data. There are some parts where a certain degree of redundancy exists but this does not hurt and may even help stress some relevant points. I have only a few comments to improve the manuscript, and I recommend a minor revision before publication.*

We thank Dr. Daniele Penna for this positive assessment and his thoughtful comments, which we have addressed in detail below.

*Comments of the reviewer are shown in italics.* Responses from the authors are presented in regular font below each comment. Citations from the manuscript are in Times New Roman and changes of manuscript text are underlined.

**Specific comments**

*P1L9-10, and P1L18. Here, and in other parts of the manuscript, I suggest specifying that this work focuses on the two-component hydrograph separation, which is used to estimate the pre-event and event component of stream runoff.*

Thank you for pointing this out, we will change that.

*Indeed, the word "source", used, for instance, at P1L18, P2L17 is, in my opinion, a bit vague: one tracer (often isotopes), two-component hydrograph separation is typically used to estimate \*time\* source components of total discharge, whereas two-tracer (or more than two tracers, usually isotopes plus hydrochemical tracers), three (or multi-) component hydrograph separation is often used to estimate \*geographical\* source components (Klaus and McDonnell, 2013), such as snowmelt, glacier melt, or hillslope soil water, riparian soil water, shallow groundwater etc. I think that this specification and distinction should be made clear in the abstract and throughout the paper (not only in the title of Section 2.4).*

We will change that: "Tracer-based, two-component hydrograph separation uses stable water isotopes ($^2$H, $^{18}$O) to estimate the relative time source components of streamflow, i.e. recent precipitation (event water, $Q_e/Q$) and catchment storage (pre-event water, $Q_{pe}/Q$; Klaus and McDonnell, 2013). "

*P3L10. This is true and was shown in several studies. However, also the opposite can happen, depending on the specific catchment properties. For instance, in our Dolomitic experimental catchment tracer data showed that, on average, Qe increases with increasing antecedent moisture conditions mainly due to the streamflow contribution of saturation overland flow, formed by a mixture of rain water falling on the saturated areas and pre-event water exfiltrating in the riparian zone (Penna et al., 2016). This may occur also in other catchments.*

*Penna, D., van Meerveld, H.J., Zuecco, G., Dalla Fontana, G., Borga, M., 2016. Hydrological response of an Alpine catchment to rainfall and snowmelt events. Journal of Hydrology 537, 382–397. https://doi.org/10.1016/j.jhydrol.2016.03.040*

We will include this reference into the revised version of the manuscript: "In contrast, at sites where positive relationships between $Q_e/Q$ and antecedent wetness have been observed, it has been hypothesized that vertical infiltration must first replenish storage deficits before event water can be rapidly transported via lateral flow pathways or surface runoff (Shanley et al, 2002; von Freyberg et al., 2017), or that the expansion of saturated areas in the catchment enhances direct runoff of rainwater (Penna et al., 2016)."

*P4L15-20. The general aim of the study is clear but I suggest formulating here specific objectives and/or a clear testable hypothesis.*

We will re-phrase this section: "From the general concepts outlined above, we hypothesize that event and pre-event runoff coefficients $Q_e/P$ and $Q_{pe}/P$ may be more informative metrics for studying catchment storm responses, compared to the widely used event water fraction of discharge $Q_e/Q$ or the runoff coefficient $Q/P$. In this paper, we test this hypothesis by comparing runoff coefficients $Q/P$ and the different ratios $Q_e/Q$, $Q_{pe}/Q$, $Q_e/P$ and $Q_{pe}/P$ across 24 storm events and analyzing their relationships with storm characteristics and antecedent moisture. These relationships shed light on possible streamflow generation processes at our study site and highlight the potential benefits of using $Q_e/P$ and $Q_{pe}/P$ rather than $Q/P$ or $Q_e/Q$ to characterize catchment storm response."

*P7, Section 2.3. I think that the selection of metrics used to characterize the storm properties and the antecedent wetness conditions are appropriate. However, I think it would be interesting to add the combination of P and SMini as a metric (e.g.,Detty and McGuire, 2010; Fu et al., 2013) and see if and how the fractions of Qe/Q, Qpe/Q, and most of all Qe/P and Qpe/P are sensitive to it.*

*Detty JM, McGuire KJ. 2010. Threshold changes in storm runoff generation at a till-mantled headwater catchment. Water Resources Research 46: W07525. DOI:10.1029/2009WR008102*

*Fu C, Cheng J, Jiang H, Dong L. 2013. Threshold behavior in a fissured granitic catchment in southern China: (1) analysis of field monitoring results. Water Resources Research 49: 1–17. DOI: 10.1002/wrcr.20191*

We agree that the combined metric of *SMini* and *P* might provide an interesting analysis, however, in our case the event-to-event variations in *P* are much larger than those in *SMini*. Therefore, the correlations with the combined metric (*P+SMini*) are very similar to those obtained for *P*.

P18L11. I agree but AP7 (and AP indices in general) is only a surrogate of the catchment antecedent wetness status (Ali and Roy, 2010) and therefore this relation could not be robust and reliable "fingerprint". Maybe a sentence on this could be added.

Ali, G. A. and Roy, A. G.: A case study on the use of appropriate surrogates for antecedent moisture conditions (AMCs), Hydrol. Earth Syst. Sci., 14, 1843-1861, https://doi.org/10.5194/hess-14-1843-2010, 2010.

We will add this information to acknowledge the comment of the reviewer: "Under the assumption that $AP7$ is a reliable surrogate for catchment antecedent moisture, the slope of the $AP7$-$Q_{pe}/P$ relationship could be considered as an index of how antecedent moisture alters the fraction of the catchment in which stored, pre-event water can be efficiently mobilized by streamflow. "

**Minor comments and technical corrections**

*P1L9. It is not immediately clear if the terms "streamflow" and "discharge" are used interchangeably or if they imply a different meaning. In the first case, I suggest to use one term consistently. In the second case, I suggest to indicate the possible distinction.*

In our analysis, the term "streamflow" usually refers to the hydrological behavior or status of a system (i.e., the streamflow regime, the streamflow hydrograph), whereas "discharge" refers to the variable $Q$ that is used for our conceptual and theoretical explanations. We will clarify this terminology throughout the manuscript.

*P5L3. I suggest replacing "soil depths are shallower" with "soils are shallower".*
We will change that

*P8L6. So, in the last 2.5 hours?*
Not necessarily. If a drift control was measured before the beginning of an event, the time interval becomes 3 hours.

*P10L21. Please, report the p-value here as well.*
The p-value was <0.0001 in all cases. We will add this information.

*P12L7. "river": earlier in the manuscript the authors used the term "streamwater" (eg, P7L17), so I imagine (also considering the catchment size) that the term "stream" is more appropriate here.*
We will correct that.

*P14L2-3. As far as I understand, the authors here mean "but are not, however, statistically significant (p>0.01)" or "but are, however, statistically not significant (p>0.01)".*
Thank you for catching this error. We will correct that.

*P14L26. I suggest adding a reference here. Examples might be McGlynn and McDonnell (2003), James and Roulet (2009), Muñoz-Villers and McDonnell (2012).*
*James AL, Roulet NT. 2009. Antecedent moisture conditions and catchment morphology as controls on spatial patterns of runoff generation in small forest catchments. Journal of Hydrology 377(3-4): 351–366. DOI: 10.1016/j.jhydrol.2009.08.039*
*McGlynn, B. L., and J. J. McDonnell (2003), Quantifying the relative contributions of riparian and hillslope zones to catchment runoff, Water Resour. Res., 39, 1310, doi:10.1029/2003WR002091, 11.*
*Muñoz-Villers LE, McDonnell JJ. 2012. Runoff generation in a steep, tropical montane cloud forest catchment on permeable volcanic substrate, Water Resources Research 48: W09528. DOI: 10.1029/ 2011WR011316*
These studies did not explicitly analyze the relationships of antecedent wetness metrics to $Qpe/P$, which is discussed in this section. Therefore, these references are not directly relevant here.

*P14L28. "which sum to the runoff coefficient itself: Q/P=Qe/P+Qpe/P". This has been said more than once before, and can be dropped.*
We will change that.

*P15L3. I suggest replacing "tightly" with "strongly".*
We will change that.

*P18L18. I suggest replacing "forever" with "also for large values" or something similar.*
We will change that: "We also note that  these four  ratios are logically constrained to be ≤1, and thus they must become asymptotic at some point."

*Table 2. I suggest dropping the second and the third column (Q and P) because already reported in Table 1. This can improve the readability of the table.*
We agree and will change Table 2 accordingly.

*Fig. 2. The label of the last panel should be "d)" and not "e)".*

We will correct that.

*Fig. 3. So, if I understand well, this flow duration curve is a combination of two distinct periods. I wonder whether it would be more appropriate to show two curves for the two periods separately.*

Since we do not consider the two periods separately in the analysis, we would like to refrain from showing two flow duration curves.

*Fig. 4. In the label of the two y-axis correct "18Q" with "18O".*

Thank you for catching this error, we will correct Fig. 4.

*Fig. 7. The caption is not complete.*

Sorry, the last two lines of the caption ended up on the next page of the automatically-generated PDF.

---

## Author Comment (AC2) · 11 Oct 2018

**Response to the interactive comment of Reviewer#2**

on "Studying catchment storm response using event and pre-event water volumes as fractions of precipitation rather than discharge" by Jana von Freyberg et al.

*Best authors and editors,*

*Thank you for the possibility to review this very interesting manuscript, and apologies for the delay in my review.*

*The authors present a hydrograph separation studying the stream water sources in an experimental Erlenbach catchment in Switzerland. The work builds on an advanced field laboratory, enabling high-frequency determination of isotope composition in stream water and precipitation used in identifying pre-event and event water composition, respectively. Authors present an eight-month long dataset of isotope and hydrometric measurements for flow and precipitation, supplemented with groundwater level and soil moisture data as proxies for catchment wetness. As a subset of this data, they analyse 24 storms in greater detail. The results show the advantages in exploring the pre-event and event water contributions as a fraction of precipitation, not total streamflow as is typically done. Using this approach, the authors were able to infer novel insights to catchment controls on streamflow generation. I particularly enjoyed section "3.4 fingerprints of catchment response" in which the authors put forward interesting hypothesis to be tested by the hydrological community.*

*The manuscript is written with flawless English, and is well structured and presented. In my opinion both the collected and dataset and the following analysis are novel and of high quality, and therefore a great contribution to the hydrological sciences. I recommend this work to be published in HESS, and provide some minor remarks below.*

We thank the reviewer for his/her assessment and his thoughtful comments, which we have addressed in detail below.

*Comments of the reviewer are shown in italics.* Responses from the authors are presented in regular font below each comment. Citations from the manuscript are in Times New Roman and changes of manuscript text are underlined.

**comments:**

*P4L13: I would recommend the authors to better acknowledge and discuss prior work studying the Qe/P ratio in the introduction. Before this chapter, I had the impression this is being done the first time in the presented manuscript.*

Indeed, except for the one study cited here (Ocampo et al., 2006), all other studies estimated Qe/P solely as a proxy for surface-runoff generating area. Nonetheless, in the revised version we will mention these prior studies earlier in the introduction, in the sentence after we introduce Qe/P. To the best of our knowledge, none of these studies, including Ocampo et al.'s study, used Qe/P the way we did, and therefore this is being done here for the first time.

*P5L5 what do you mean by "saturated soils"? groundwater table is at ground level? Or that the soil type is prone to saturation? I presume that the extent of saturation would vary seasonally, so a static map for it seems simplified.*

Soil surveys across the catchment landscape revealed that the soils are frequently saturated and this can be caused by both shallow groundwater tables or waterlogging of oncoming rainfall on low-permeability soils. Both processes are likely to co-occur and are difficult to separate (Fischer et al., 2015; Rinderer et al., 2017). We agree with the reviewer that a static map of soil wetness is a simplified description of the catchment wetness state, however, no data are available about the spatiotemporal variability of these areas.

What has been mapped in our study catchment, as we say clearly in the text, are zones where soil saturation is likely to occur, rather than locations that are saturated at any specific point in time.

Fischer, B. M. C., Rinderer, M., Schneider, P., Ewen, T., and Seibert, J.: Contributing sources to baseflow in pre-alpine headwaters using spatial snapshot sampling, Hydrol. Process., 29, 5321-5336, 10.1002/hyp.10529, 2015.

Rinderer, M., McGlynn, B. L., and van Meerveld, H. J.: Groundwater similarity across a watershed derived from time-warped and flow-corrected time series, Water Resour. Res., 53, 3921-3940, 10.1002/2016WR019856, 2017.

*Fig.1: add a scale, the degree axis are not very intuitive of the catchment size*
We will add a scale bar to Fig. 1.

*P7L16: concentrations -> ratios?*
*We will change that: "*We use the isotopic composition ($\delta^2$H and $\delta^{18}$O) of streamwater and precipitation*"*

*P8L10: how is Q for each event defined and calculated?*
We have moved the definitions of the start and end times of the event from Section 2.3 to Section 2.4*: "*The beginning of a storm event was the time of first rainfall, and the end of a storm event was defined as the time that (i) event water discharge declined to 5 % of its peak value or (ii) another precipitation event began, whichever came first; case (i) prevailed for 18 of 24 events.*"*  Q for each event is the sum of discharge between these beginning and ending times.

*P10L1: add spacing for dates in all occurrences?*
We will follow whatever format specification the journal requires, although we would like to keep this format as we use it here because it provides concise identifiers for the individual events.

*P10 L15: I don't understand how the 4-hour peak Q 0.11 mm is lower than overall Q 0.5 mm. How is 4-hour peak Q defined?*
These are cumulative sums, not rates (that is: total mm, not average mm/hr).  4-hour peak flow ($Q_{4h}$) is defined as the cumulative sum of discharge volume over a 4-hour time period. We will add an explanation to Section 2.3.  Since the aggregation period is usually much longer for total cumulative discharge $Q$ (aggregated over the entire event duration), $Q_{4h}$ is smaller than $Q$.

*Fig. 4: should y-axis be delta 18O?*
We will correct that.

*P10L 24: how about the point on the far right in both a) and b) plots? That deviates substantially from the 1:1 line.*
For the event with the largest $Q_e/Q$ values (18Aug2017), the uncertainties of the hydrograph separation with $\delta^{18}$O and $\delta^2$H were relatively large and thus the differences in the $Q_e/Q$ values (i.e., the deviation from the 1:1-line) were statistically not significant (in the specific sense that they did not differ by more than twice their pooled standard errors).  We will revise the text to make this clear.

*P15L10 and table3: I don't find Qpe/Q data in table 3, though discussed in the text*
Because Qpe/Q + Qe/Q =1, the correlation of Qpe/Q with anything will be simply the negative of the correlation of Qe/Q.  We point this out several times in the paper, e.g. in the introduction (P4L5) and discussion (P14L10-11). We will add this information also to the legend of Table 3 and the caption of Fig. 7.

*P17L6: I see this conclusion somewhat inconsistent with your data analysis so far. You suggest that the Pe could be explained by contraction and expansion of saturated areas, i.e. the antecedent conditions, whereas before you demonstrate and discuss how the Pe is mainly a function of the storm characteristics.*
We are not sure whether the reviewer refers to Qe, Qpe or the ratios Qe/P? Regarding P17L6, which discusses Qe/P, we want to point out that some of these saturated areas form on impermeable surfaces or on waterlogged soils, which remain hydrologically isolated from the groundwater aquifer or the stream network most of the time.  Therefore, the expansion and contraction of saturated areas in Erlenbach does not necessarily need to be reflected in the metrics of antecedent wetness conditions (e.g., Qini, GWini).

---

## Referee Comment (RC3) · Anonymous Referee #3 · 13 Oct 2018

General comments: The authors propose a new approach to characterize the catchment response of a pre-alpine mountainous catchment Erlenbach, in Switzerland. They measured high-resolution precipitation and stream flow isotopic data to calculate pre-event and event water fractions based on precipitation instead of discharge, as commonly used. A large number of storm events (24 events, in total) are analysed combined with, for example, antecedent moisture conditions in the catchments. The results shown in this study underline their potential to a new "fingerprint" of catchment responses. With respect to the transferability of these results, it is clear that a cross comparison study is needed, as it is already mentioned in this study. However, I recommend to extend the remarks on whether this fingerprinting approach might hold also

in other catchments and to hypothesize which requirements would need to be fulfilled (catchment characteristics or climate such as mountainous, specific land cover proportion, temperature climate and so on). Besides, the manuscript is written in a concise way and in good English quality. Some figures deserve further attention regarding additional information in the caption or their readability. To conclude, only minor revision is needed for this study to be accepted in Hydrology and Earth System Sciences.

Specific comment: Page 1, Line 14: At this point, it is not clear for the reader if event-averaged fractions or instantaneous fractions are used. Please clarify. Page 8, Line 5: Is it correct that the average isotopic composition is taken from the period of time of 2,5h prior to the storm event? This information could be added here in parenthesis, for example. Page 10, Line 3: If your data analysis starts 6 May 2017, the effect of snowmelt on the isotopic composition of the stream water is still present and should be considered when discussing May and June storm events (for example, 13 and 19 May 2017 events). Both events occur after intense snowmelt infiltration into the stream (see Figure 2). Page 10, Line 10: Which technical problems occurred with the automatic sampler? However, it is not necessary to report these details in the manuscript. Page 14, Line 2-3: Please remove the first "not". Page 18, Line 4: Qini in Figure 7 is not displayed in log-scale Page 18, Line 16-18: Please discuss further whether these 'fingerprint' may results from the specific catchment characteristics of Erlenbach catchment and how strongly they are connected to the catchment land cover.

Table 1: the last date entry of column 1 is '29Ot2017'. Figure 1: Please enlarge map symbols and make labels more visible (using a different colour and fontsize, for example). Please correct 'The Erlenbach...' in the caption. Figure 2: Although these events are not considered for your analysis, please mention the remaining events, during which isotopic stream composition remarkably drops (snowmelt events?) Figure 3:On which criteria is the selection of events displayed here based? Please report. Figure 5: Could you add errorbars in this graph? Figure 7: Axis labels and tick labels are very small and difficult to ready. Please enlarge here to improve readability.

---

## Author Comment (AC3) · 15 Oct 2018

**Response to the interactive comment of Reviewer#3**

on "Studying catchment storm response using event and pre-event water volumes as fractions of precipitation rather than discharge" by Jana von Freyberg et al.

**General comments:**

*The authors propose a new approach to characterize the catchment response of a pre-alpine mountainous catchment Erlenbach, in Switzerland. They measured high-resolution precipitation and stream flow isotopic data to calculate pre-event and event water fractions based on precipitation instead of discharge, as commonly used. A large number of storm events (24 events, in total) are analysed combined with, for example, antecedent moisture conditions in the catchments. The results shown in this study underline their potential to a new "fingerprint" of catchment responses. With respect to the transferability of these results, it is clear that a cross comparison study is needed, as it is already mentioned in this study. However, I recommend to extend the remarks on whether this fingerprinting approach might hold also in other catchments and to hypothesize which requirements would need to be fulfilled (catchment characteristics or climate such as mountainous, specific land cover proportion, temperature climate and so on). Besides, the manuscript is written in a concise way and in good English quality. Some figures deserve further attention regarding additional information in the caption or their readability. To conclude, only minor revision is needed for this study to be accepted in Hydrology and Earth System Sciences.*

We thank the reviewer for his/her assessment and his thoughtful comments, which we have addressed in detail below.

*Comments of the reviewer are shown in italics.* Responses from the authors are presented in regular font below each comment.  Citations from the manuscript are in Times New Roman and changes of manuscript text are underlined.

**Specific comments:**

*Page 1, Line 14: At this point, it is not clear for the reader if event- averaged fractions or instantaneous fractions are used. Please clarify.*

The ratios $Q_e/Q$, $Q_{pe}/Q$, $Q_e/P$, $Q_{pe}/P$, and $Q/P$ were calculated from the cumulative volumes of event water (Qe), pre-event water (Qpe), total discharge (Q=Qe+Qpe) and precipitation (P) aggregated over the storm period. Instantaneous values were denoted with an index i, as pointed out in Sect. 2.4.

We will be more explicit in the revised Abstract to emphasize that our analysis is based on cumulative volumes (mm) and not on instantaneous rates (mm/h).

*Page 8, Line 5: Is it correct that the average isotopic composition is taken from the period of time of 2,5h prior to the storm event? This information could be added here in parenthesis, for example.*

Yes, this is correct. However, if a check standard was sampled within this period, the time span became 3 hours.  We will add this information in Sect. 2.4 in the revised manuscript.

*Page 10, Line 3: If your data analysis starts 6 May 2017, the effect of snowmelt on the isotopic composition of the stream water is still present and should be considered when discussing May and June storm events (for example, 13 and 19 May 2017 events). Both events occur after intense snowmelt infiltration into the stream (see Figure 2).*

We agree with the reviewer that the more isotopically depleted streamwater in early-mid May might reflect snowmelt contributions.  However, to the extent that snowmelt inputs will contribute to both pre-event water and event water, they are not likely to exert a substantial effect on the hydrograph separations.  This is particularly the case because the isotopic composition of incoming rainwater during those May and June events used in our analysis was very distinct from that of pre-event water stored in the catchment (see detailed time series plots in the Supplement Material), and the bulk snowpack sampled in the catchment (which will be presented in an upcoming paper).  Measurements from snowmelt lysimeters (which will also appear in an upcoming paper) also show that the isotopic composition of the water leaving the snowpack

during these events was dominated by recent precipitation (in the upper part of the catchment, which still had some snow cover in May). Thus, despite the strong seasonality in the streamwater isotopic composition, we believe that the two-component hydrograph separation results provide reasonable estimates of event- and pre-event water volumes.

*Page 10, Line 10: Which technical problems occurred with the automatic sampler? However, it is not necessary to report these details in the manuscript.*

The automatic sampling routine was programmed in the MagICNet software of the ion chromatograph (IC). Thus whenever there was a technical problem with the IC, the sampling routine (which also fed the isotope analyzer) was interrupted (von Freyberg et al., 2017).

von Freyberg, J., Studer, B., and Kirchner, J. W.: A lab in the field: high-frequency analysis of water quality and stable isotopes in stream water and precipitation, Hydrol. Earth Syst. Sci., 21, 1721-1739, 10.5194/hess-21-1721-2017, 2017.

*Page 14, Line 2-3: Please remove the first "not".*

Thank you for catching this error. We will correct that.

*Page 18, Line 4: Qini in Figure 7 is not displayed in log-scale*

We will update the tick marks of the $Q_{ini}$-axis in the revised version of the manuscript.

*Page 18, Line 16-18: Please discuss further whether these 'fingerprint' may result from the specific catchment characteristics of Erlenbach catchment and how strongly they are connected to the catchment land cover.*

Because our data set was collected at only one catchment, we should not speculate about the role of the catchment characteristics in the fingerprints discussed here. For this, a similar analysis needs to be carried out at sites with different landscape and climatic properties – which we point out at the end of the paragraph.

Synthetic results from a benchmark model show that different parameter values result in different fingerprints (Kirchner, 2018), suggesting that the relationships observed for Erlenbach might not be universally applicable, and, on the other hand, that these fingerprints may vary substantially in response to variations in site characteristics. But because we have real-world data from only one site, we cannot yet say how Erlenbach's "fingerprints" might compare to those observed elsewhere.

Kirchner, J.: Quantifying new water fractions and transit time distributions using ensemble hydrograph separation: theory and benchmark tests, Hydrol. Earth Syst. Sci. Discuss., https://doi.org/10.5194/hess-2018-429, in review, 2018.

*Table 1: the last date entry of column 1 is '29Ot2017'.*

The last entry was actually '29Oct2018', which we will correct to '29Oct2017'.

*Figure 1: Please enlarge map symbols and make labels more visible (using a different colour and fontsize, for example). Please correct 'The Erlenbach. . .' in the caption.*

We will change that.

*Figure 2: Although these events are not considered for your analysis, please mention the remaining events, during which isotopic stream composition remarkably drops (snowmelt events?)*

These are not snowmelt events. As explained in the text, they were not analyzed because there was not a clear isotopic separation between the event and pre-event water, so the event water contribution could not be reliably determined using hydrograph separation.

*Figure 3: On which criteria is the selection of events displayed here based? Please report.*

We picked 16 events with the most diverse peak flow rates. For the sake of readability, the remaining eight events are not included in the graph because they share very similar hourly peak flow rates to those events depicted in Fig. 3. We will add this information to the caption of Fig. 3.

*Figure 5: Could you add errorbars in this graph?*

Due to the high-resolution isotope data, the standard errors of the cumulative volumes of Qe and Qpe are very small – typically much less than one mm (Table 2). Therefore, the error bars in Fig. 5 would often be too small to be visible. We will include an information about the relative errors of Qe and Qpe in the caption of Fig. 5.

*Figure 7: Axis labels and tick labels are very small and difficult to read. Please enlarge here to improve readability.*

We will change that.